# Human cancer-targeted immunity via transgenic hematopoietic stem cell progeny

Theodore S. Nowicki [1,2,3,4,5,6] ✉, Nataly Naser Al Deen[7], Cole W. Peters [1], Begoña Comin-Anduix[3,8,9], Egmidio Medina [7], Cristina Puig-Saus [2,3,4,9,10], Ignacio Baselga Carretero[7], Paula Kaplan-Lefko[7], Mignonette H. Macabali [7], Ivan Perez Garcilazo[7], Daniel Chen [7], Jia Pang[7], Beata Berent-Maoz[7], Salem Haile[3,6,7], Jonathan Rodriguez [6], Moe Kawakami [1], Conner K. Kidd [1], Ameya Champhekar[7], Giuseppe Carlucci[11], Agustin Vega-Crespo[7], Bartosz Chmielowski [3,7], Arun Singh[7], Noah Federman[1,3,4,6,12], Gary M. Schiller [3,7], Sarah J. Larson[3,7], Martin Allen-Auerbach[6,8], Alexandra M. Klomhaus [13], Jerome Zack[2,7], David Baltimore [14], Lili Yang[2,3,4,5], Donald B. Kohn [1,2], Owen N. Witte[2,3,4,5,6,10] & Antoni Ribas [3,4,6,7,9,10,11]

Adoptive transfer of genetically engineered T cells expressing a tumor-antigen-specific transgenic T cell receptor (TCR) can result in clinical responses in a variety of malignancies. However, these responses are frequently short-lived, and patients typically relapse within several months. This phenomenon is largely due to poor persistence of the transgenic T cells, as well as a progressive loss of their functionality and terminal differentiation in vivo. This underscores the need for cell therapy approaches able to sustain the initial antitumor efficacy and lead to long-term antitumor efficacy. Herein, we report the use of tandem cell therapies involving autologous T cells and hematopoietic stem cells engineered to express the NY-ESO-1 TCR for the treatment of solid tumors in a first-in-human phase I clinical trial (NCT03240861). This therapy is shown to be safe, feasible, and leads to initial tumor regression activity. T cell progeny from the HSC progenitors is shown to provide circulating transgenic NY-ESO-1 TCR-T cells, which display tumor-antigen-specific antitumor functionality, without any evidence of anergy or exhaustion. These results demonstrate the utility of transgenic HSCs to generate a self-renewing source of tumor-specific cellular immunotherapy in human participants. Clinicaltrials.gov: NCT NCT03240861

Immunotherapy with T cells genetically engineered to express a tumor antigen-specific transgenic T cell receptor (TCR) has enabled the treatment of some solid tumors, mainly metastatic sarcoma, melanoma, and human papilloma virus (HPV) positive cancers[1–6]. This therapeutic approach has been shown to induce objective tumor regression in a large proportion of patients. However, these initial responses are often not sustainable, and patients frequently relapse within 6–12 months following the initial cell infusion[1–6]. This phenomenon underscores the need for better cell therapy approaches to sustain the antitumor efficacy and lead to durable disease control and remission. Previous work by our group and others has shown that transgenic cell therapies are often associated with decreases in transgenic T cell persistence over time[3,4,7]. Furthermore, the persisting transgenic T cells themselves often acquire a hypofunctional terminal

effector and exhausted phenotype, which coincides with their loss of therapeutic efficacy and cancer progression[3,8]. These phenomena collectively underscore the need for therapeutic approaches that can overcome these weaknesses by enhancing the persistence of transgenic T cells with highly functional antitumor activity in order to provide sustained therapeutic efficacy.

Given the weaknesses in traditional modalities of transgenic TCR-T cell therapy, we developed a novel approach to these treatments. Previous studies have shown that TCR-transduced hematopoietic stem cells (HSCs) endogenously differentiate into fully active mature T cells in animal models[9–13]. However, there is a delay in the appearance of such T cells in peripheral circulation. Therefore, we designed a phase I clinical trial for patients with advanced solid tumors expressing the tumor antigen NY-ESO-1, a pan-cancer antigen expressed in a wide variety of solid tumors (predominantly sarcomas), but generally not within any normal somatic tissues. In the present study, in an effort to potentiate the therapeutic efficacy of traditional TCR-T cell therapeutics, we delivered a tandem adoptive cell therapy of transgenic NY-ESO-1 TCR autologous T-cells along with autologous HSCs expressing the same transgenic NY-ESO-1 TCR after a reduced-intensity myeloablative and lymphoablative chemotherapy conditioning regimen[14]. Furthermore, the HSC product was engineered to co-express a modified herpes simplex virus (HSV) thymidine kinase 1 (sr39TK), which allowed for both the non-invasive tracking of progeny derived from gene modified peripheral blood stem cells (PBSC) to assess their biodistribution as well as functioning as a suicide gene to ablate the transgenic cells in the event of unanticipated toxicity[13,15].

In the present study, we report that tandem therapy with both transgenic T-cells and HSCs encoding the NY-ESO-1 TCR is safe, feasible, and provides an initial wave of antitumor activity arising from the traditional TCR transduced T cell product, after which the T cell progeny from the HSC niche are shown to provide circulating transgenic NY-ESO-1 TCR T cells which display specific NY-ESO-1 antigen-dependent antitumor functionality, the first such instance of this approach being successfully utilized in human subjects. We also report the first successful use of the suicide/reporter gene sr39TK to detect and monitor in vivo engraftment of transgenic hematopoietic stem cells in human subjects. These results have implications for the design of future iterations of transgenic cell therapy for the treatment of solid tumors and other conditions.

## Results

### Patient enrollment, manufacture, and administration of transgenic TCR peripheral T-cell and hematopoietic stem cell products

Patients with relapsed or refractory metastatic solid tumors and the following criteria were considered for enrollment in the trial: NY-ESO-1-positive tumors as demonstrated by immunohistochemistry; HLA-A:02:01 status as demonstrated by high-resolution molecular phenotyping; disease progression with standard-of-care treatment options for their metastatic or locally advanced cancer; adequate clinical performance status; and evidence of measurable disease and fulfilling eligibility criteria. These participants were considered for an initial mobilized leukapheresis at UCLA, during which they received filgrastim (granulocyte colony-stimulating factor) and plerixafor (anti-CXCR4) to mobilize and collect PBSCs. CD34+ cells were isolated using automated magnetic separation and cell sorting and were then resuspended in serum-free medium with supplemental cytokines SCF, FLT3L, TPO, and IL-3. The cells were cultured for $18 \pm 6$ h prior to transduction with a clinical-grade lentiviral vector encoding the NY-ESO-1 TCR and the HSV-sr39TK gene. This latter feature allowed for both non-invasive in vivo imaging of the transgenic HSCs via 9-4-[18 F] fluoro-3-(hydroxymethyl)butylguanine ([18 F]-FHBG) positron emission tomography–computed tomography (PET/CT)[16–22], as well as a potential target for ablation in the event of any toxicities resulting

from the HSC product[13]. Following lot release testing of the transduced HSC product, patients underwent a non-mobilized leukapheresis to collect peripheral blood mononuclear cells (PBMCs), and were admitted to the hospital for conditioning chemotherapy with 70,000–80,000 ng/mg*hr of busulfan and 40 mg/m2/day of fludarabine (days −5 thru −2)[23]. In parallel, their PBMCs were transduced with a retroviral vector encoding the NY-ESO-1 TCR. Transgenic PBSCs were infused on day 0, and transgenic PBMCs were infused on day +1, followed by low-dose IL-2 (500,000 IU/m2) administered subcutaneously twice daily on days +2 through +8 as tolerated. Manufacturing protocols and cell infusion timelines are summarized in Fig. 1. Five patients enrolled in the clinical trial, and three received both cellular therapy infusions; their manufacturing and therapeutic outcomes are summarized in Table 1 and Supplementary Tables 1 and 2. Two patients who had consented did not receive the double cellular therapy: subject NYSCT-02 was unable to mobilize sufficient CD34+ PBSCs despite repeated mobilization attempts, while subject NYSCT-04 developed a malignant pleural effusion requiring indwelling chest tube for drainage shortly before his mobilized leukapheresis, and withdrew from the study.

### Assessment of safety, feasibility, and efficacy

There were no grade 3 or greater adverse events attributed to the autologous NYESO-1 TCR/sr39TK transgenic HSCs, while one grade 3 adverse event (hypotension) was graded as possibly related to the retroviral modified NYESO-1 TCR PBMCs. The investigational cell therapy agents were both able to be administered safely and feasibly. Adverse events are summarized in Supplementary Data 1. While there were multiple grade 4 adverse events attributed to the conditioning chemotherapy and resultant immunosuppression, there were no grade 3 or greater adverse events attributed to the autologous NYESO-1 TCR/sr39TK transgenic HSCs. The incidence of adverse events relating to conditioning chemotherapy agents (busulfan and fludarabine) was consistent with the significant and prolonged pancytopenias, which are often associated with the use of these agents in the setting of autologous HSC transplant procedures and adoptive cell therapy protocols. Of the patients who ultimately received therapy, two (NYSCT-03 and NYSCT-05) demonstrated reduced tumor size in response to therapy (Supplementary Fig. 2), while one patient did not show an objective clinical response to therapy (NYSCT-01).

Subject NYSCT-03 developed Grade 4 hypoxia on day +62, which was found to be due to cytomegalovirus (CMV) pneumonitis. Subject was CMV-negative by serology pre-treatment, and had persistently negative CMV serum PCRs post-treatment, so this was deemed to represent a de novo infection and not a reactivation. Subject's CMV was refractory to treatment with multiple antiviral agents, and she passed away on day +76. Subject NYSCT-05 experienced Grade 3 malignant pleural effusion requiring chest tube placement and admission to the hospital for observation on day +92, which was determined to be related to disease progression.

### Visualization of transgenic HSC niche in vivo correlates with persistence of lentiviral progeny in circulation

Peripheral blood samples were obtained weekly through day +28, and biweekly thereafter. Isolated PBMCs were monitored for transgenic TCR-T cell persistence via flow cytometry and interrogation for both retroviral and lentiviral vector presence via ddPCR. Differences between viral vectors allowed delineation between TCR-T cells that were derived from the initial PBMC infusion product or from the transgenic HSC niche (Supplementary Fig. 3). Patients underwent PET scans with the experimental tracer [18 F]-FHBG on approximately day +30 and day +120 in order to visualize the transgenic HSCs and correlate with the findings in peripheral circulation. The sr39TK protein phosphorylates an experimental PET reporter probe, [18 F]-FHBG, upon IV administration. [18 F]-FHBG, a radiolabeled analogue of

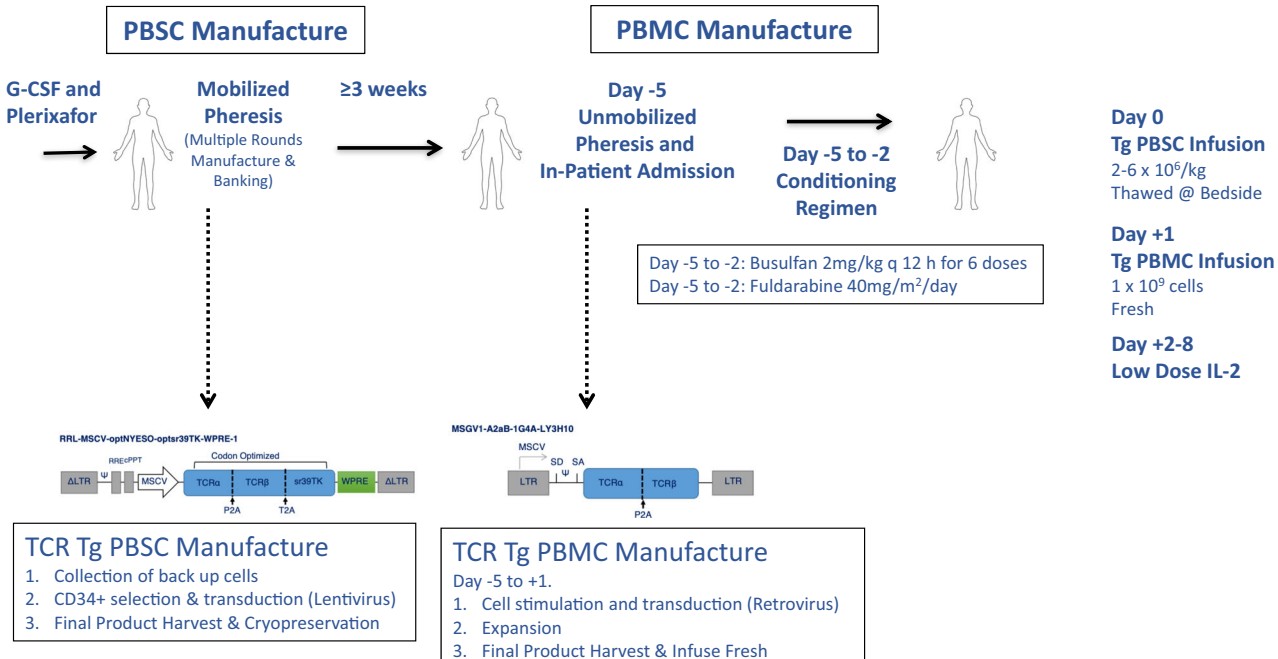

**Fig. 1 | Overview of manufacturing protocol, and clinical trial procedures.**
Peripheral blood stem cells (PBSCs) were isolated from patients via leukapheresis following bone marrow mobilization with filgrastim and plerixafor. CD34+ cells are isolated via magnetic bead separation using the CliniMACS platform, and were then transduced with the lentiviral vector encoding the NY-ESO-1 TCR and sr39TK reporter/suicide gene (enabling non-invasive visualization of the transgenic HSC progeny in vivo within the bone marrow niche, as well as serve as a safety ablation feature), and were then cryopreserved. Following final lot release criteria for the gene-modified PBSCs, patients were admitted following unmobilized leukapheresis to collect peripheral blood mononuclear cells (PBMCs), which were then transduced with a retroviral vector encoding the NY-ESO-1 TCR while the patient received conditioning chemotherapy with busulfan and fludarabine in parallel, in order to selectively myeloablate and lymphodeplete the patient, respectively. Gene-modified PBSCs and PBMCs were administered to the patient on day 0 and day +1, respectively, and patients subsequently received 7 days of IL-2 for transgenic T-cell expansion in vivo. Created in BioRender. Nowicki, T. (2025) https://BioRender.com/4bcin2w.

penciclovir, becomes trapped in the sr39TK-expressing PBSCs and progeny T cells upon phosphorylation and is detectable by PET imaging[24]. 7–7.5 mCi of [18 F]-FHGB was injected for each PET/CT scan for determination of sr39TK transgenic HSCs. The normal biodistribution of [18 F]-FHBG does not demonstrate significant uptake in bone marrow or splenic tissue[24]. Visual assessment was used for evaluation of the Day +30 and Day +120 post-transplant scans available for the patients. In addition, volume of interest (VOI, 1 cm³) of representative marrow (L5 vertebra) and splenic regions was placed, and the maximum standardized uptake value ($SUV_{max}$) and $SUV_{peak}$ were calculated for all scans (Supplementary Table 3). Subject NYSCT-03 passed away prior to the day +120 timepoint, while subject NYSCT-05 displayed no engraftment at day +30, so no day +120 scan was performed on either of these subjects.

Although each subject demonstrated different success rates with the engraftment of the transgenic HSCs, these results were consistent with our overall hypothesis regarding the generation of functional lentiviral TCR-T cell progeny from the HSC niche. NYSCT-01 demonstrated only very modest expansion of the transgenic TCR-T cells (Fig. 2A, B), which was attributed to suboptimal lymphodepletion with fludarabine (30 mg/m2/day on days −4 and −3), given that the busulfan is specifically myeloablative for CD34+ HSCs. The protocol was therefore modified, and the next patients NYSCT-03 and NYSCT-05 received higher doses of fludarabine, resulting in superior peripheral lymphodepletion (Supplementary Fig. 4) and subsequent in vivo expansion of the transgenic TCR-T cells. The post-transplant [18 F]-FHBG PET scans for NYSCT-01 demonstrated initially robust engraftment and consequent visualization of the lentiviral transgenic HSCs by day +29, but showed loss of engraftment at the next scan on day +120 (Fig. 2C, D). Consistent with this finding, NYSCT-01 demonstrated an initially retroviral-predominant signature in PBMCs (owing to the

retroviral TCR-T cell infusion), which was then superseded by lentiviral-predominant PBMC signatures coinciding with the engraftment of lentiviral-transduced HSCs and their corresponding progeny (Fig. 2A, B). However, lentiviral progeny were subsequently undetectable by day +120, which was consistent with the loss of transgenic HSCs noted at that point by the corresponding [18 F]-FHGB PET. Of note, no patients' PBMCs demonstrated significant increase in auto/allo-reactivity (as measured by interferon-gamma release assay) following coculture with their own PBSCs due to the presence of the sr39tk gene (Supplementary Fig. 5). Conversely, subject NYSCT-05 showed robust expansion of TCR-T cells in vivo (Fig. 3A), which was an initially retroviral-predominant PBMC population, but never displayed any detectable lentiviral progeny within the PBMC niche (Fig. 3B). Consistent with this finding, no detectable engraftment at day +35 as visualized via [18 F]-FHGB PET scan for NYSCT-05 (Fig. 3C). This was attributed to the borderline transduction efficiency of that patient's transgenic HSC preparation, although this was still within the lot release criteria parameters (Supplementary Table 1). Consistent with this finding,

## TCR transgenic progeny from genetically modified HSC niche demonstrates antigen-specific anti-tumor functionality

NYSCT-03, like NYSCT-01, demonstrated an initially retroviral-predominant signature within the expanding transgenic TCR PBMC precursors, which was superseded by lentiviral signature concomitantly with the engraftment of transgenic HSCs (Fig. 4). However, the PBSC transplant within patient NYSCT-03 engrafted and showed continued expansion up to day +43 after transplantation, unlike NYSCT-01 whose PBSC engraftment had begun to decline after day +30. To further assess the functional engraftment of the lentiviral cell product, we subjected the patient's day +43 PBMCs to joint single

**Table 1 | Patient demographics and outcomes**

| Patient study number | Sex (M/F) | Ethnicity | Age | Type of Cancer | Prior Treatments | Active Disease Sites | Stage | Evidence of transient tumor response | Response at day 60 | PFS (mo) | OS (mo) | Sites of Progression | Post-Study Treatments |
|---|---|---|---|---|---|---|---|---|---|---|---|---|---|
| NYSCT-01 | M | Hispanic | 25 | Epitheloid Sarcoma | Pazopanib | Right forearm; right humerus; pulmonary nodules | IV | No | PD | 2 | 20 | Liver | Trabectedin, pazopanib |
| NYSCT-02 | F | Caucasian | 21 | Synovial Sarcoma | Radiation; doxorubicin+ifosfamide; enoblituzumab+doxorubicin; high-dose ifosfamide; anlotinib; regorafanib | Right chest wall; pulmonary nodules | IV | NA (did not ultimately receive treatment/ withdrew from study) | NA | NA | NA | NA | NA |
| NYSCT-03 | F | Caucasian | 28 | Synovial Sarcoma | Surgery; Radiation; doxorubicin+ifosfamide; olaratumab+doxorubicin; gemcitabine+docetaxel; pazopanib | Right thigh; pulmonary nodules | IV | Yes by PET/CT | PR | 2.5 | 2.5 | NA | NA |
| NYSCT-04 | M | Hispanic | 56 | Synovial Sarcoma | Radiation; Doxorubicin+ifosfamide; high-dose ifosfamide; anlotinib; gemcitabine+docetaxel +ADI-PEG20; Trabectedin+nivolumab +T-VEC | Right hand; pulmonary nodules | III | NA (did not ultimately receive treatment/ withdrew from study) | NA | NA | NA | Left pectoralis mass | Ifosfamide, nivolumab, Yondelis |
| NYSCT-05 | M | Caucasian | 38 | Synovial Sarcoma | Doxorubicin+ifosfamide; anlotinib | Right popliteal fossa; Lung | IV | Yes by PET/CT | SD | 2 | 10 | Intrathoracic LN and pulmonary nodules | INY |

*F* female, *LN* lymph nodes, *M* male, *NA* not applicable, *PFS* progression-free survival, *OS* overall survival, *EOS* end of study, *mo* months.

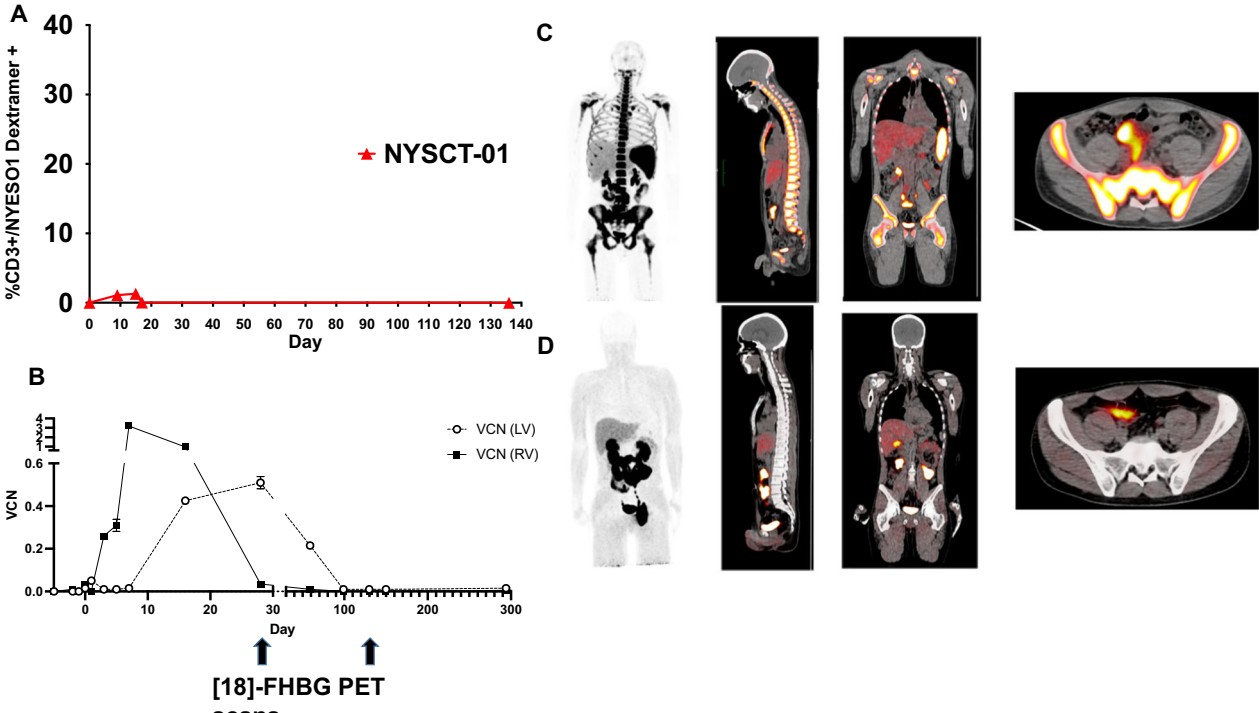

**Fig. 2 | Retroviral and lentiviral-transduced cell engraftment kinetics in patient NYSCT-01. A** Peripheral NY-ESO-1 TCR+ expansion and contraction over time in vivo. **B** Vector copy number (VCN) kinetics in peripheral circulation. While there was a predominance of retrovirus (RV)-derived product early, there was predominantly lentiviral (LV)-derived product coinciding with the day +29 [18 F]-FHBG PET (**C**), followed by a loss of LV-derived product coinciding with the day +120 PET scan (**D**). Data in 2B represent two experimental replicates, ± SD.

nuclei RNA sequencing (snRNAseq) and assay for transposase-accessible chromatin (ATACseq), which allows to determine chromatin accessibility across the genome along with the whole transcriptome sequencing from the same nucleus, we confirmed that there were very few cells containing retroviral signature (-0%), while a substantial fraction of the nucleated cells contained lentiviral signatures (46.9%), indicating that they were derived from the transgenic HSC progenitors (Fig. 5A–C). These lentiviral-positive PMBCs (pos_pLenti) represented a diverse range of nucleated myeloid and lymphoid cell types, including a range of different T cell phenotypic subtypes, as shown by the different cell types and cell states (Fig. 5D–E).

To assess whether the PBSC cell products, which give rise to T cell progeny, would ablate normal TCR expression and thus reduce T cell repertoire reconstitution, we also analyzed snRNAseq data in T cell subtypes for TCR expression. The lentiviral-positive T cells (pos_pLenti) from peripheral blood demonstrated some degree of allelic exclusion of the native TCR locus, consistent with previously reported studies of this system in murine models[25], although this phenomenon was not complete. While the lentivirus-positive CD4 T cell clusters demonstrated ~45% co-expression of both the lentiviral NY-ESO-1 TCR (pRRL-lenti) and genomic TRBC1/2, only 15.4% of CD8 T cell clusters co-expressed the lentiviral NY-ESO-1 TCR along with the native genomic TRBC2, without TRBC1. The remainder of the CD4 and CD8 T cell clusters had only lentiviral NY-ESO-1 TCR expression detected (Fig. 5F–J). Taken together, these results demonstrated that the transgenic lentiviral TCR-T cell progeny were able to arise from the transgenic HSCs, as we had initially hypothesized, with some incomplete degree of allelic exclusion demonstrated by co-expression of transgenic and genomic TCRs within the T cell clusters.

Of note, focusing on the pLenti+ and pLenti- CD8 T-cell populations showed no difference in anergy and exhaustion markers expression in the CD8 T-cells across the pLenti+ and pLenti- cells (Supplementary Fig. 6A, B). Looking into the top 100 differentially expressed genes between pLenti+ and pLenti- CD8 T-cells did not reveal key transcriptional differences between the two CD8 T-cell populations. Although CD80 appeared to be upregulated in a few pLenti+ CD8 T-cells (only 3 out of 24 cells) as compared to the pLenti- CD8 T-cells, we suspect that the pLenti+ CD8 T-cells are new progeny that wouldn't have had the chance to undergo thymic selection (Supplementary Fig. 6C).

Finally, we sought to assess the antigen-dependent functionality of these lentiviral HSC-derived TCR-T cells. When we stimulated these cells from the day +43 PBMC sample from NYSCT-03 with the human melanoma cell line M257-A2, which is NY-ESO-1-positive and HLA-A:02:01-positive, we found significant elevations in secreted interferon-gamma, compared to the T cells cultured alone or with M257 cells, which are positive for NY-ESO-1 but negative for HLA-A:02:01 (Supplementary Fig. 7). Taken together, these results demonstrated that the transgenic lentiviral TCR-T cell progeny arising from the transgenic HSCs displayed antigen-specific cytokinetic functionality, consistent with our fundamental hypothesis of the mechanism of action of this therapy.

## Discussion

Transgenic adoptive cell therapy approaches for solid tumors are an effective form of cancer immunotherapy, with a wide variety of targets. However, their lack of persistent anti-tumor activity stands in stark contrast to their frequent and initially potent early objective clinical responses, including in recently FDA-approved cell therapies for solid tumors, such as afamitresgene autoleucel[26]. This widespread finding underscores the need to improve the persistence of the early anti-tumor activity of the transgenic T cells. Given that our group and others have shown that the persistence and progressive terminal differentiation of the adoptively transferred T cells coincides with the loss

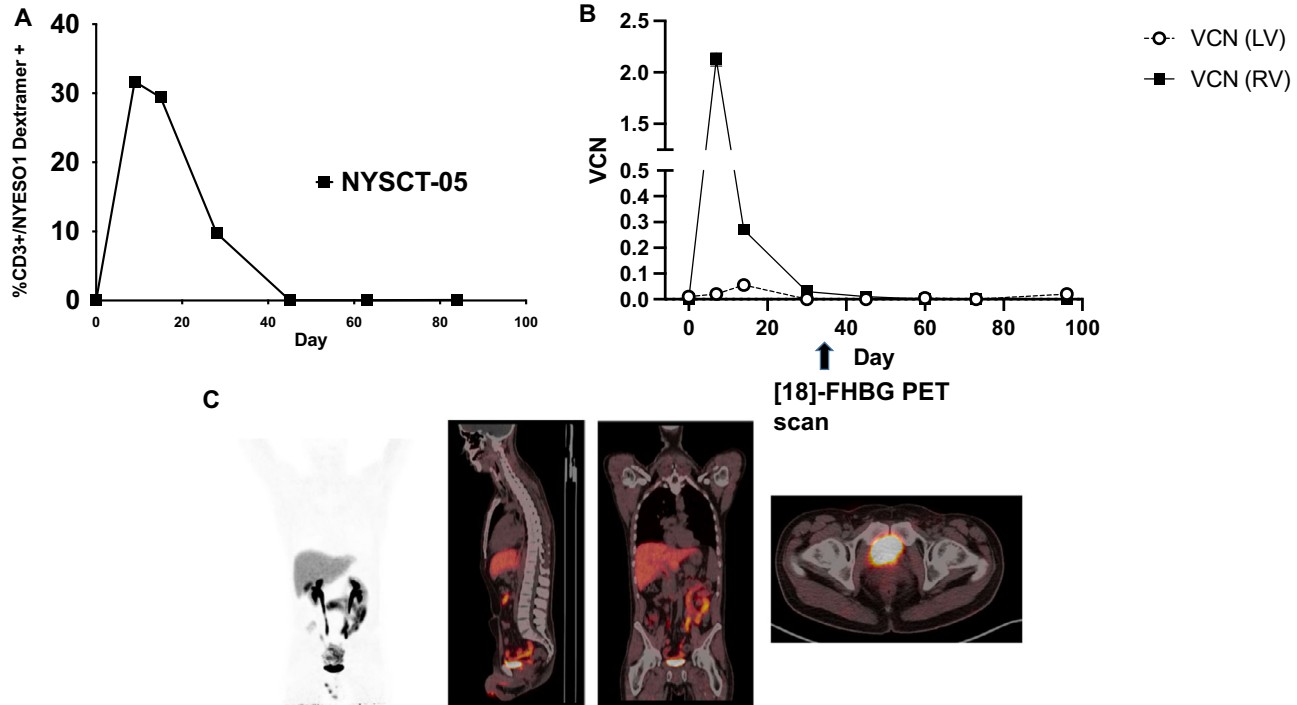

**Fig. 3 | Retroviral and lentiviral-transduced cell engraftment kinetics in patient NYSCT-05. A** Peripheral TCR+ expansion and contraction over time in vivo. **B** Vector copy number (VCN) kinetics in peripheral circulation. While there was a predominance of retrovirus (RV)-derived product early, there was no significant detectable lentiviral (LV) progeny at any point, consistent with the non-engraftment observed which was observed at the day +35 [18 F]-FHBG PET scan (**C**). Data in 3B represent two experimental replicates, ± SD.

of their therapeutic efficacy[3,8,27], along with the previously demonstrated ability of transgenic TCR HSCs to generate transgenic progeny in peripheral circulation in murine models[9,10,12], we sought to test this approach in a first-in-human trial with a tandem transfer of a transgenic HSC product along with the traditional transgenic TCR-T cell product.

The ability of the transgenic HSC precursors to generate progeny containing the NY-ESO-1 TCR vector insert is consistent with previous studies incorporating this technique to replace missing or deficient enzymes in the setting of primary immunodeficiencies. In such studies, the transgene was present in all nucleated progeny arising from the transgenic marrow and demonstrated functional persistence of expression for years following administration of the autologous gene-modified product[28,29]. Lentiviral vectors utilized in this capacity have also demonstrated a degree of persistence and safety superior to that of retroviral vectors[30–33], which have been associated with both transformation-induced malignancies[31,34] and epigenetic silencing of the retroviral vector's expression[35,36]. While using this approach to both generate functional TCR transgenic T cells arising from the HSC niche and utilizing the sr39TK gene as an in vivo visualization has been shown in animal models[9,10,12], this represents the first time that either technique has been effectively demonstrated in human subjects. Our single-cell data showed the expected presence of the transgenic TCR within all lineages interrogated; however, it is unlikely that any expression of TCR protein in a non-T cell would lead to any antigen-specific activities, given that these cells would lack the other underlying intracellular components necessary to effectuate a TCR-driven response.

Given the long-term persistence and functionality of other lentiviral-modified HSC products' transgenes, it is possible that our approach would have resulted in the sustained generation of functional transgenic TCR-T cells capable of engaging with the NY-ESO-1 tumor antigen. Although previous studies have cast doubt on the

ability of HSC-derived CAR-T cells to survive thymic selection[37], both our current data and our previously published experiences with NY-ESO-1 TCR-T cell therapies have demonstrated that most subjects have low-level, but detectable levels of circulating NY-ESO-1 TCR clones prior to TCR transduction[3,38]. This implies that NY-ESO-1 is sufficiently antigenic as to generate TCR clones which survive thymic selection, albeit with a target which is insufficiently antigenic as to generate a robust clonal response. Given that we are using such a TCR as is present in endogenous settings, we would speculate that this would be a potential advantage in using a TCR instead of a CAR. However, the logistical considerations of this therapy, coupled with the inherent clinical precariousness of patients with aggressive sarcomas, limited our ability to examine how these products would have gone on to equilibrate long-term, and may ultimately limit the wider therapeutic efficacy of this approach.

The difficulty in both initiating and maintaining the therapeutic efficacy of this therapy, even in the context of demonstrated functional progeny arising from the HSC niche, underscores the inherent risks of an autologous stem cell transplant as a component of refractory sarcoma treatment. While autologous stem cell transplant has been utilized in the setting of high-dose chemotherapy consolidation (with autologous stem cell rescue) for high-risk Ewing's sarcoma and desmoplastic small round blue cell tumors, these approaches are associated with significant toxicities beyond those encountered with conventional chemotherapy[39,40]. Given the death of subject NYSCT-03 due to a de novo CMV infection during immunologic reconstitution (despite the generation of functional transgenic T cell progeny from the transgenic HSCs), as well as the evidence of partial allelic exclusion in the patient's T cells arising from the transgenic HSCs, the risks inherent to our approach may ultimately outweigh the therapeutic benefit. Indeed, we were also prevented from obtaining on-treatment tumor biopsy samples for analysis from this patient owing to her

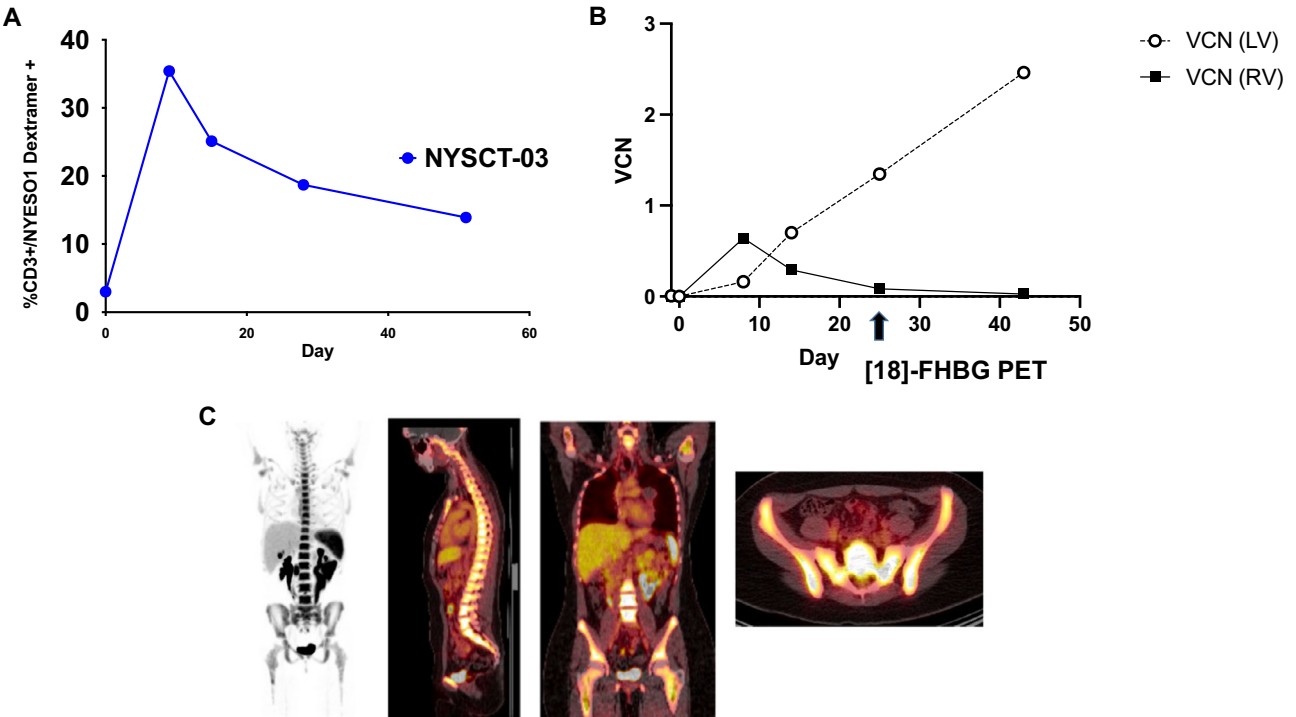

**Fig. 4 | Retroviral/lentiviral engraftment kinetics in patient NYSCT-03.**
**A** Peripheral TCR+ expansion and contraction over time in vivo. **B** Vector copy number (VCN) kinetics in peripheral circulation. While there was a predominance of retrovirus (RV)-derived product early, there was predominantly lentiviral (LV)-derived product coinciding with the day +25 [18 F]-FHBG PET (**C**). There was no data beyond day +43, as the patient had passed away, and there was insufficient sample at day +53 for analysis. Data in 4B represent two experimental replicates, ± SD.

clinical instability following her initial tumor regression, which also limited our ability to derive additional conclusions regarding the in vivo activity of the T-cell progeny derived from the transgenic HSCs. Furthermore, the inability of subject NYSCT-02 to mobilize sufficient CD34+ PBSCs, as well as NYSCT-04 withdrawing prior to mobilization due to rapid disease progression, underscores the burden of both heavy pre-treatment chemotherapy and the inherent medical fragility of patients with relapsed/refractory sarcoma. Given that our approach required a 28-day wash-out period, as well as several weeks to allow for the transgenic PBSC manufacture, the logistics of a treatment modality involving transgenic HSCs may preclude the treatment of medically fragile patients with heterogeneous relapsed or treatment-refractory sarcomas, further limiting its broader clinical utility.

The reason for the initial engraftment and subsequent loss of the transgenic HSCs in subject NYSCT-01 was unclear, although possibilities include poor long-term persistence of the transgenic HSCs, or T cell-mediated rejection of the transgenic marrow. If the latter possibility were the mechanism of graft loss, the sr39TK reporter/suicide gene could be a potential cause for this. While the potential immunogenicity of the native HSV-TK protein has been shown in T cell products[41,42] it has not been shown to occur in the context of using sr39TK in HSC transduction[12]. Furthermore, our own co-culture experiments with patient-matched PBMCs with transduced and untransduced PBSCs did not result in any significant reactivity as measured by interferon-gamma release.

Despite the clinical difficulties in utilizing a stem-cell-based therapeutic in patients with advanced sarcomas, this represents the first instance of successfully generating functional clonal transgenic T cell progeny from lentiviral-transduced HSCs. The implications for such success can be applied to a multitude of other diseases. The use of genetically modified HSCs has been proposed to generate antigen-specific T cells directed against HIV-1 infection, and has been validated

in preclinical models[43–45]. Furthermore, the use of this approach to generate long-term chimeric antigen receptor (CAR) T cell or invariant natural killer T (iNKT) cell progeny has also been explored in preclinical models[46–48]. It is possible that the patient populations of other potential disease indications, such as those with HIV-1 infection, would not have the burdens of heavy chemotherapy pre-treatment history or rapidly progressive malignancy, and could be more suitable for such genetically engineered HSC-based immunotherapies than our target patient population. Furthermore, the use of a CAR construct might avoid the partial allelic exclusion phenomenon observed both in preclinical studies utilizing a TCR-transduced HSC therapy, as well as our own. However, the unknown impact of such CAR expression on all HSC-derived progeny underscores the need for continued use of suicide gene systems as a safety feature in any such future studies.

In conclusion, the administration of dual cell therapy with retroviral transgenic TCR-T cells and lentiviral transgenic TCR HSCs in three subjects was safe and feasible, with no significant adverse events attributed to the cell therapy products. Visualization of transgenic HSCs via [18 F]-FHGB in human subjects was effective in tracking successful engraftment. Lentiviral TCR-T cell progeny arising from the transgenic HSC niche demonstrated NY-ESO-1 antigen-specific cytokinetic functionality, as well as a significant degree of allelic exclusion. Points for improvement of this therapeutic modality include improvements in transduction efficiency of the lentiviral HSC product, mitigation of risks from opportunistic infections, and subject selection, given the medical fragility of patients with sarcoma and the time needed to enroll and treat patients with this therapy. Given the preclinical data demonstrating the use of gene-modified HSCs to generate CAR-T cell or iNKT cell products, as well as TCRs directed against non-cancer antigens, such as HIV-1, future iterations of this immunotherapy modality could be expanded to include a diverse range of therapeutic indications.

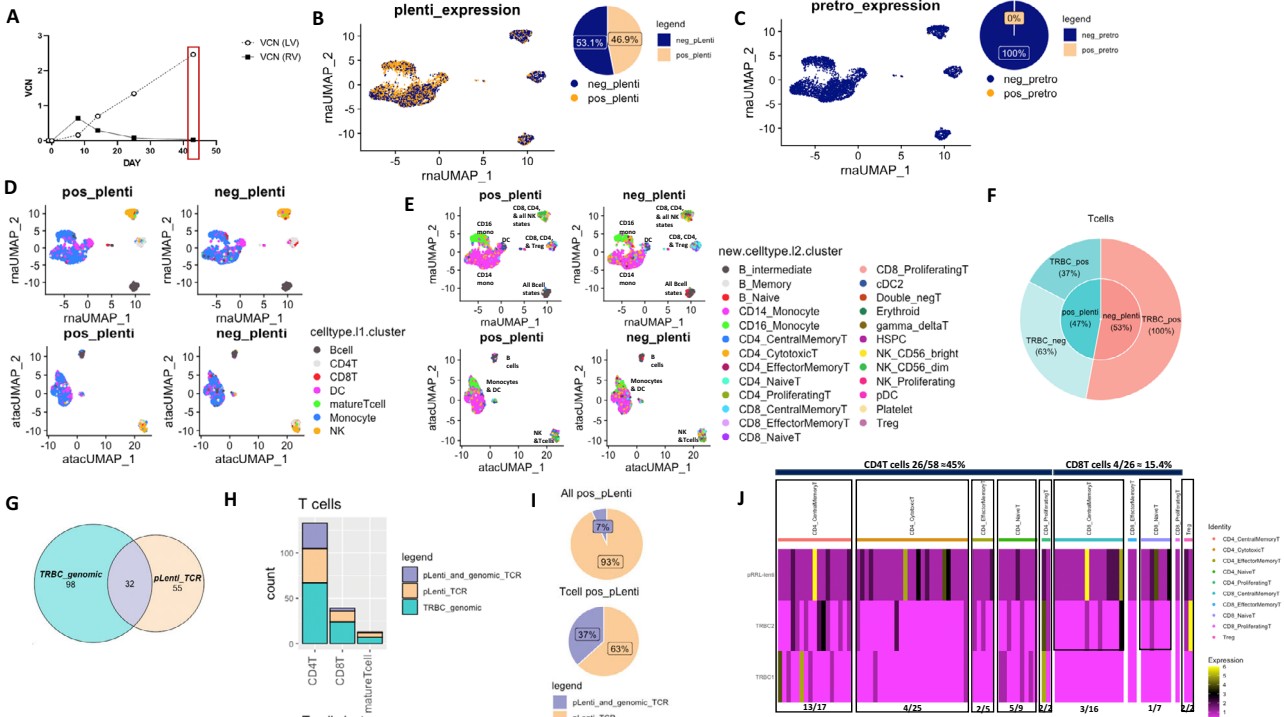

**Fig. 5 | Single nuclei RNA sequencing (snRNAseq) of PBMC from patient NYSCT-03 at day 43. A** Overview of ddPCR VCN kinetics at timepoint selected for analysis, which yielded 4166 nuclei after doublet removal and QC (as shown in Fig. 4B). Of the PBMC population, 1954 nuclei were lentiviral-positive (pLenti + , 46.9%) and 2212 nuclei were lentiviral-negative (pLenti-, 53.1%) (**B**), while only 10 nuclei were retroviral-positive (pretro + ) and 4156 were retroviral-negative (pretro-) (**C**). Both pLenti+ and pLenti- cells expressed all immune cell types from Azimuth References at comparable levels in RNA and ATAC UMAPs (**D**), and all immune cell type states at comparable levels; myeloid cells clustered together, B cells clustered separately, and T cells and NK cells clustered together in RNA and ATAC UMAPs (**E**). NYESO1 TCRB lentivirus encodes the same amino acids as T cell receptor beta (TRBC1) gene, but is codon optimized. T cells represented 185 nuclei (≈4.5%) of all nuclei, which were equally represented by pLenti+ and pLenti- nuclei (47% and 53%, respectively). Of pLenti+ Tcells, 37% co-expressed genomic TRBC1/2 + codon optimized lentiviral NY-ESO-1 TCR (pRRL-lenti), while 63% exclusively expressed codon optimized pRRL-lenti TCR (**F**). Of the T cells, 32 nuclei co-expressed the genomic TRBC1/2 + the pLenti TCR, while 98 and 55 nuclei had mutually exclusive expression of either genomic TRBC1/2 or codon optimized pRRL-lenti TCR, respectively (**G**). T cell CD4:CD8 ratio was 3:1, and the proportion of nuclei belonging to CD4 T cell, CD8 T cell, or other mature T cells was comparable for nuclei expressing exclusively TRBC1/2 (blue) or nuclei expressing codon optimized pRRL-lenti TCR (**H**). Of all pLenti+ nuclei (1954), 7% co-expressed genomic TRBC1/2 (131), while of all pLenti+ T cells (87 nuclei), 37% co-expressed genomic TRBC1/2 (32 nuclei) (**I**). Heatmap for expression of pLenti TCR, TRBC2, and TRBC1 per nucleus for pLenti+ T cells revealed allelic exclusion in the majority of nuclei. 45% of the CD4 T cell clusters/state co-expressed the NYESO1 pLenti TCR and genomic TRBC1/2, while 15.4% of the CD8 T-cell cluster/states co-expressed NYESO1 and genomic TRBC2 but not TRBC1, remaining nuclei exclusively expressed the codon optimized NYESO1 pLenti TCR (**J**).

## Methods

### Study ethics and conduct

Patients were enrolled in the clinical protocol after signing a written informed consent approved by the UCLA Institutional Review Board (#15-000511) under investigational new drug designations for the NY-ESO-1 TCR (IND#17471 and #15167) and the 9-4-[18 F]fluoro-3-(hydroxymethyl)butylguanine ([18 F]-FHBG) PET tracer (IND# 135251). The study was conducted in accordance with local regulations, the guidelines for Good Clinical Practice, and the principles of the Declaration of Helsinki. The study had the clinical trial registration number NCT03240861, and this is the final report of this study. Use of human specimens from this trial was also approved by the UCLA Institutional Review Board under the above protocol number.

### Trial eligibility and screening procedures

Eligible patients were HLA-A*0201 by high-resolution molecular phenotyping, with locally advanced or metastatic solid tumors, and with either no available standard therapeutic options with curative intent, or progression on standard-of-care chemotherapy or radiotherapy regimens. Tumors were NY-ESO-1 positive by immunohistochemistry. Patients were >18 years of age, with a life expectancy of >3 months, an ECOG performance status of 0 or 1, adequate organ function required to receive systemic IL-2[49], and seronegative for Hepatitis B/C, and HIV.

Patients with clinically active brain metastases were excluded. More than four weeks had elapsed since any prior systemic steroid use or cancer therapy.

### Cell manufacturing processes and treatment schedule

The manufacture and treatment schedule is outlined in Fig. 1. Patients underwent mobilized leukapheresis for the manufacture of the NY-ESO-1 TCR/sr39TK transgenic PBSCs. Patients self-administered filgrastim (Neupogen, Amgen, Thousand Oaks, CA; 10 μg/kg/day) for four days prior to the leukapheresis, and a 20 mg dose of plerixafor (Mozobil, Sanofi, Paris, France) the evening before leukaphersis. A minimum of $5 \times 10^6$/kg body weight of CD34+ cells, as determined by clinical flow cytometry, was required prior to initiating manufacturing procedures.

Following harvest, CD34+ HSCs were purified, cultured, and transduced in the UCLA Good Manufacturing Practices facility, using standard CliniMACs® cell purification protocols and seeded into fresh closed cell culture bags or flasks in serum free medium (X-VIVO-20, Lonza, Basel, Switzerland) supplemented with 1% human serum albumin (Baxter), and cytokines SCF (300 ng/mL), Flt-3 ligand (300 ng/mL), TPO (100 ng/mL) and IL-3 (20 ng/mL) at a density of 0.5 to $1 \times 10^6$/mL. All cytokines (Peprotech, Cranbury, NJ) and culture reagents were approved for ex vivo clinical use. A small culture sample was removed

for sterility testing. After ~18 h total pre-stimulation, cells were cultured with the NY-ESO-1 TCR/sr39TK LV (Supplementary Fig. 1A, Supplementary Data 2) (Indiana University Vector Production Unit, Indianapolis, IN) for ~20 h in the same complete medium at a cell concentration of 0.5 to $1 \times 106$ cells/mL. Following transduction, a portion of cells were removed for quality control i.e., analysis of transduction efficiency (via flow cytometry, colony-forming assay, and ddPCR), viability, sterility, and replication-competent lentivirus. Full details of in-process testing and lot release testing are available in Supplementary Table 4 and Supplementary Data 3. Following lot release testing, the remaining cells were washed and resuspended in CryoStor® CS5 (BioLife, Bothell, WA) for cryopreservation prior to administration.

Patients underwent an unmobilized leukapheresis for PBMC isolation for the manufacturing of the NY-ESO-1 TCR transgenic T cells, which were manufactured as previously described[3]. Briefly, unselected PBMC were isolated by Ficoll gradient centrifugation and cultured in AIM V media (Gibco, Invitrogen, Chicago, IL) supplemented with 5% human AB serum (Omega Scientific, Tarzana, CA) in the presence of 50 ng/mL anti-CD3 (OKT3, Miltenyi Biotec, Auburn, CA) and 300 IU/mL IL-2 in order to stimulate T cell growth to prepare for viral vector transduction. Activated PBMC were infected by the clinical grade MSGV1-A2aB-1G4A-LY3H10 retroviral vector (Supplementary Fig. 1B and Supplementary Data 4) supernatant using retronectin-coated plates (NYSCT-01) or bags (NYSCT-03, NYSCT-05) (Retronectin, Takara, Otsu, Shiga, Japan) for two consecutive days, maintained in culture for 4 days from the start of transduction in IL-2, and then infused fresh into trial subjects. Aliquots of these cells were used to fulfill the lot release criteria. Full details of in-process testing and lot release testing are available in Supplementary Table 5 and Supplementary Data 3.

Patients were admitted immediately following unmobilized leukapheresis, and received conditioning chemotherapy with busulfan 70,000–80,000 ng/mg*hour (days -5 thru -2) and fludarabine 40 mg/m²/day (days -5 thru -2)[23]. On Day 0, patients received the transgenic PBSC infusion (thawed at bedside and then administered with 500 mL normal saline). On day +1, patients received up to $10^9$ of the transgenic TCR lymphocytes as an IV infusion of freshly prepared cells. Patients then received low-dose IL-2 therapy (500,000 IU/m2 subcutaneously) the following morning (day +2) twice daily for up to 14 doses (through day +8), as tolerated. Standard supportive care provided post-infusion included filgrastim, antibiotics for neutropenic fever, twice weekly CMV and EBV serum PCRs, and blood product transfusions as needed. When patients recovered peripheral neutrophil counts and were no longer transfusion-dependent, they were discharged from the hospital. A research [18F]FHBG positron emission tomography-computed tomography (PET/CT) scan with IV contrast was performed on day +30 as described below, and formal restaging scans (CT or MRI, as clinically indicated) were performed on day +60.

### Safety assessments
Adverse events were analyzed following NCI CTCAE v3.0. The known toxicities/side effects of the preparative chemotherapy, IL-2, filgrastim, or plerixafor, as listed in the protocol or package insert, were not considered for the assessment of DLTs. A GALV quantitative PCR (qPCR) assay, developed and optimized at the IU VPF[50], was used to rule out the presence of replication-competent retrovirus (RCR) and replication-competent lentivirus (RCL). This assay was performed on infusion cell products as well as retrospectively in PBMC processed from post-infusion peripheral blood samples at 3, 6, and 12 months. Annual samples were archived thereafter.

### Assessment of feasibility
After entering two patients in each cohort, followed for a minimum of one month after the last subject had received the infusion of the transgenic PBSCs and PBMCs, an assessment of protocol feasibility was performed by the study investigators. Both cell therapies required strict lot release criteria of the final product before administration (as discussed above). Feasibility was based on potential problems in the manufacturing of either cell product or potential problems in the delivery of either cell product in the clinical protocol. If two patients could not receive the intended cellular therapies, further accrual would not be warranted.

### Assessment of antitumor activity
Quantification of changes in PET imaging for the intratumoral accumulation of [18F]FDG within tumor sites and secondary lymphoid organs was performed by counting the total number of FDG avid lesions, as well as the maximum standardized uptake value (SUVmax), normalized to the body weight of the patient at baseline and on day +60. The objective clinical response rate was assessed on day +60 scans and recorded following modified Response Evaluation Criteria in Solid Tumors (RECIST1.1[51]).

### MHC dextramer immunologic monitoring
Detection of NY-ESO-1 TCR expression using fluorescent MHC dextramer analysis for NY-ESO-1 (Immudex, Copenhagen, Denmark) was performed on cryopreserved peripheral blood mononuclear cells (PBMCs) collected at different time points, as previously described in refs. 4,52. Our previous definitions for a positive or negative immunologic response using standardized MHC multimer assays were used, which are based on the assay performance specifications by defining changes that were beyond the assay variability with a 95% confidence level[52]. Representative flow cytometry plot is shown in Supplementary Fig. 8.

### In vivo imaging of transgenic HSCs via [18 F]-FHBG PET
Patients were injected with 200 MBq [18F]FHBG approximately 25 days and 120 days after ACT, with allowed variability of ± 10 days and ±30 days, respectively. One-hour post injection, a PET/CT scan was acquired. Scans of the extremities were performed in patients with metastatic lesions in the arms or legs. The CT scans were performed in a "low dose mode" (110 kVp, 30 mAs anode current), resulting in an effective dose of less than 1.3 mSv per scan[53,54]. These doses are comparable with typical diagnostic CT or PET scans and below the limits set by federal regulations for research studies involving radioactive imaging agents. The US Food and Drug Administration has previously approved its use as an investigational new drug in clinical trials[55].

Regional uptake of [18F]FHBG within metastatic tumor sites and secondary lymphoid organs was quantified by SUV normalized to the body weight of the patient. As an internal quality control, SUVs were also be determined for several normal organs, such as muscle, liver, and lungs. These measurements allowed identification of any technical problems in the SUV calculations, such as partially paravenous tracer administration[56,57]. All imaging studies were performed on the same PET/CT system in order to eliminate confounding effects of differences in scanner sensitivity, spatial resolution, or image processing.

### Digital droplet PCR to determine retroviral and lentiviral progeny persistence in vivo
DNA from transduced and mock PBMC/T-cells was isolated via the AllPrep DNA/RNA Micro Kit (Qiagen, Hilden, Germany). ddPCR was carried out using the ddPCR Supermix for probes (Biorad 1863023, Hercules, CA), using probes and specific primers to differentiate the lentiviral or retroviral NY-ESO1 TCR sequence. For lentivirus or retrovirus detection, a common forward primer was used LVorRV R: GTCTCTCAGCTGGTACACGGC with either lentiviral specific LV F: TAAGAGCAGCGGCCGGTCC, or retroviral specific RV F: GCTGGA-TAAATCATCAGGACGTAGT, in combination with genomic calibrator SDC4 gene primers F: CAGGGTCTGGGAGCCAAGT, R: Probe:

GCACAGTGCTGGACATTGACA. PCR was run and VCN determined by ddPCR as per manufacturer instructions. FAM conjugates probes were used to detect viruses; Lentiviral probe: ACCAGCCTGATCGTG-CACCCCTACA, and retroviral probe: AACCAGCCTTATTGTT-CATCCGTAT. SDC4 calibrator probe was conjugated to HEX: CCCACCGAACCCAAGAAACTAGAGGAGAAT.

## Antigen stimulation co-culture experiments

For antigen stimulation co-culture experiments, patient NYSCT-03 PBMC samples were thawed and recovered overnight in RPMI 1640 (Thermo A1049101, Waltham, MA) supplemented with heat-inactivated 10% fetal Bovine serum (R10) and IL-2 to a final concentration of 17 ng/mL (BioLegend #589102, San Diego, CA) (R10 + IL2). The next day PBMCs were plated with adherent human melanoma cells M257 (which are positive for NY-ESO-1), or with M257 cells which were transduced to express HLA-A:02:01 (M257-HLA) cell lines[58] at a 1:1 cell ratio, and co-cultured in R10 + IL2. M257 cells were isolated and generated by Antoni Ribas as previously described[58]. Cells were tested annually for mycoplasma contamination via MyoALERT kit (Lonza, Basel, Switzerland). For ELISA experiments, melanoma cells were seeded at $1 \times 10^5$ or $5 \times 10^5$ cells per well, and then co-cultured with equal amount of PBMC cells in R10 + IL2 as above. Cells were co-cultured as described above for 20 h, afterwards, plates were centrifuged and supernatant collected. ELISAs were performed by incubating 50 µL of co-culture supernatant with biotinylated IFNγ antibody as described by the manufacturer (Invitrogen EHIFNG, Carlsbad, CA). ELISA plates were washed and incubated with secondary HRP-conjugated streptavidin, and visualized using a TMB substrate and read on a Cellular Technology Limited plate reader (Kennesaw, GA). NYSCT patient PBMC, untransduced PBSCs, and PBSCs transduced with NYESO1TCR/sr39TK LV were recovered overnight in R10 ( IL2 17 ng/mL) at concentration of $1 \times 10^6$ cells/mL. Co-cultures were set up in round-bottom 96-well plates, with $1 \times 10^5$ cells from each group or media plated in a total of 100 µL of R10 (+Il2), then incubated for 20 h. IFNy ELISA (R&D Systems, cat#DIF50C) was performed per manufacturer instructions as above. Briefly, co-culture supernatant was used in a 1:5 dilution in sandwich ELISA for human IFNy and pg/mL calculated based of standard curve (from 15 to 1000 pg/mL).

## Statistical analysis

Graphing and statistical analyses were performed using GraphPad Prism V.9. Comparisons between interferon-gamma release from subject PBMCs and transduced vs untransduced PBSCs, and from NYSCT03 co-cultures with different HLA-A*02:01 positive cells utilized the unpaired $t$-test with Welch's correction. $p$ values of <0.05 were considered significant for all analyses.

## Single nuclei suspension preparation from PBMCs

Nuclei Isolation for Single Cell Multiome ATAC + Gene Expression Sequencing Protocol from 10x Genomics (CG000365 - Rev C, Pleasanton, CA) was followed for primary cells/fragile cell thawing and nuclei isolation according to manufacturer's protocol. Briefly, NYSCT-03 patient's PBMCs from day 43 post-transduction (≈4.6 million cells) were thawed in water at 37 °C for 1–2 min. Cells were then transferred into a 50 mL conical tube, and the cryovial was rinsed with 1 mL pre-warmed media (RPMI + 10% FBS), which was added dropwise to the 50 mL tube while gently shaking at 300 rpm at 37 °C on the thermomixer. Cells were diluted by incremental 1:1 volume additions of media for a total of 5 times every minute then centrifuged at $300 \times g$ for 5 min. The supernatant was carefully removed, leaving ~1 mL to resuspend the pellet in, then 9 mL media was added, followed by centrifugation at $300 \times g$ for 5 min. The supernatant was carefully removed and pellet resuspended in 1 mL PBS + 1% BSA (wash buffer, WB), with gentle pipetting 5x. Cell suspension was transferred and then washed with 0.5 mL WB to a 2 mL microcentrifuge tube. Prior to nuclei isolation, to remove ambient/background DNA, cells were treated with DNase I (Sigma-Aldrich, D5025, St. Louis, MO), which was resuspended in 1xPBS (no Mg+, no Ca2+) to 2.7KU/mL, as an 100x stock, using 100uL into 1 mL, for a cell concentration of $1 \times 10^6$/mL. Briefly, cells were centrifuged at $300 \times g$ for 10 min at 4 °C, supernatant was removed, 300 µL DNase Solution was added, mixed 5x, incubated on ice for 5 min, followed by 2 washes by adding 1 mL WB and spun down at $300 \times g$ for 10 min at 4 °C each. Cell concentration was determined by trypan blue cell counting, and cells were then flow sorted for granulocyte removal before nuclei isolation using a 100 µm nozzle (BD FACSMelody, Franklin Lakes, NJ) using 3 µL of 7AAD in every 500 µL of cell suspension, then sorted in a 5-mL FACS tube containing 500 µL WB. Cell concentration and quality were determined by trypan blue cell counting, followed by nuclei isolation as follows. One to two million cells were added to a 2-mL microcentrifuge tube and centrifuged at $300 \times g$ for 5 min at 4 °C. The supernatant was removed and 100 µL chilled Lysis Buffer with RNAse inhibitor (Sigma Protector, 3335402001, St. Louis, MO) was added and pipetted gently 10x, incubated for 3 min on ice, then washed 3 times by adding 1 mL chilled WB for 1.5 min, mixed 5x, centrifuged at $500 \times g$ for 5 min at 4 °C, and the supernatant was gently removed. Nuclei concentration and quality were determined using Countess II FL Automated Cell Counter, and nuclei were resuspended in WB + RNAse inhibitor to a final concentration of 3230–8060 nuclei/µL. We then immediately proceeded to the 10x genomics Chromium Next GEM Single Cell Multiome ATAC + Gene Expression User Guide (CG000338) for library prep according to the manufacturer's protocol.

## 10x library preparation and sequencing of single nuclei multiome ATAC + gene expression sequencing

Chromium Next GEM Single Nuclei Multiome ATAC + Gene Expression Library & Gel Bead Kit from 10x Genomics was used according to manufacturer's protocol (CG000338 Rev F). Briefly, following nuclei isolation, transposition was performed, nuclei were counted using trypan blue, around 20,000 total nuclei were loaded, and nuclei and barcoded beads were isolated in oil droplets on the Chromium Controller instrument. This was followed by post-GEM incubation cleanup, library pre-amplification PCR, cDNA amplification, single cell ATAC library construction, and Gene Expression library construction steps according to manufacturer's protocol. Barcoded libraries were then sequenced on Illumina using half a lane of Novaseq SP 100 cycles with half a lane of XP for the Gene Expression library and one lane of Novaseq SP 100 cycles with one lane of XP for the ATAC library.

## Sequencing read alignments and quality control (QC) of Single Nuclei Multiome ATAC + Gene Expression data

Cell Ranger Arc (v2.0.2) from 10x Genomics (with Count functionality) was used for aligning reads to the human genome reference (GRCh38) with the addition of the lentivirus and retrovirus plasmids. The reference file was downloaded from the 10x Genomics website (https://support.10xgenomics.com/single-cell-gene-expression/software/downloads/latest). The custom sequences were added following 10x Genomics build a custom reference for Cell Ranger Arc (mkref) (https://support.10xgenomics.com/single-cell-multiome-atac-gex/software/pipelines/latest/tutorial/mkref). The parameters used with Count functionality include --localcores = 16 --localmem = 96.

## Single nuclei multiome ATAC + gene expression data preprocessing

For the sequenced PBMC nuclei, Seurat v.5.0.1[59] was used for all analyses. Briefly, we inputted the RNA and ATAC counts from the filtered_feature_bc_matrix.h5 output from Cell Ranger Arc (v2.0.2) from 10x Genomics and added the RNA and ATAC Assay into a Seurat object using "CreateSeuratObject" and 'CreateChromatinAssay' functions. A series of quality filters was applied to only include the barcodes that fell into the

categories recommended by Seurat: nCount_ATAC < 70000 and nCount_ATAC > 1000, nCount_RNA < 25000 and nCount_RNA > 1000, and to avoid assaying possible dead cell or a sign of cellular stress and apoptosis with too high proportion of mitochondrial gene expression over the total transcript counts, we limited the mitochondrial content percentage (percent.mt) <25. The RNA data was then scaled and normalized using Seurat's "NormalizeData" with normalization.method = " LogNormalize" and scale.factor = 10000, "FindVariableFeatures" with selection.method = "vst", and nfeatures = 2000. Data was then scaled using "ScaleData" and top 20 PCA dimensions were assayed via "FindNeighbors" and "FindClusters" (with parameters: resolution = 0.5) functions. The "RunUMAP" function (with default parameters) and the first 20 PCs were used to perform the Uniform Manifold Approximation and Projection (UMAP), a standard dimensional reduction step, to visualize the snRNA data. The filtered peak count for ATAC matrix was normalized using term frequency-inverse document frequency (TF-IDF) normalization implemented in the Signac package using "RunTFIDF" function. This procedure normalizes across nuclei, accounting for differences in coverage and across peaks, giving higher values to rarer peaks. The 'FindTopFeatures' function was used to select peaks with features with min.cutoff = "q0" for the dimensional reduction. The "RunSVD" function from Signac package was used to perform singular value decomposition on the normalized TF-IDF matrix using the selected peaks (Latent Semantic Indexing, LSI) dimension reduction. The resulting 2:50 LSI components were used for non-linear dimension reduction using the "RunUMAP" function from the Seurat package with parameter reduction = lsi.

### Single nuclei multiome ATAC + gene expression cell-type annotation

The Azimuth Reference for Human PBMC app, a reference-based mapping pipeline led by the New York Genome Center Mapping Component as part of the NIH Human Biomolecular Atlas Project (HuBMAP), was used to generate cell typing for the multiome data[60]. Briefly, the app output was inputted into our Seurat object using Seurat's "AddMetaData" function to add the predicted cell type annotation as well as the predicted cell type score for both celltype.l1 (shorter list of PBMC cell types only) and celltype.l2 (default – larger list of PBMC cell types and cell states). The prediction scores range from 0 to 1 and reflect the confidence associated with each annotation, where a prediction score > 0.75 reflects predictions supported by multiple consistent anchors. To sharpen the cell typing, we extracted the gene sets that belong to each cell type/state for each cluster from the Azimuth Reference for human PBMC, and manually added the cell type/state into Seurat using "AddModuleScore" function for each celltype.l1 and celltype.l2. We then annotated the cell type/state per nuclei based on the most enriched score across each nucleus, which resulted in a much more defined and better clustering technique.

The gene sets used for celltype.l1 and celltype.l2 (filtered for the most ubiquitously expressed genes across all nuclei to avoid overcalling) from the Azimuth Reference for human PBMC are summarized in Supplementary Data 5 and Supplementary Data 6, respectively. The genes that were used to plot the heat maps in Supplementary Fig. 7 are "RNF128", "EGR2", "EGR3", "NFATC1", "BATF", "PDCD1", "CTLA4", "LAG3", "HAVCR2", "TIGIT", "CBLB", "ITCH", "DGKA", "PTEN", "PRKAA1", "NT5E", "ENTPD1", "CD244", "IL10", "TGFB1", "SOCS1", "SOCS3" for anergy and "PDCD1", "CTLA4", "LAG3", "HAVCR2", "TIGIT", "ENTPD1", "TOX", "TOX2", "BATF", "EOMES", "TBX21", "IRF4", "PRDM1", "NFATC1", "NFATC2", "ZEB2", "GZMB", "GZMK", "GZMH", "PRF1", "CXCL13", "CXCR5" for exhaustion markers.

### Generation of coverage profiles from snATACseq and snRNAseq

Fragments from snATACseq were aligned against reference genome and filtered against barcode while lists were based on cell type of interest (e.g., Tcell+ cells). Raw fragment reads were converted into reads-per-kilobase/library-size (i.e., RPKM-like, we term this RPKB, as the final 1e6 scaling is omitted to prevent float overflow during computation). To generate coverage of the fragments across an area of interest (Supplementary Fig. 9A), we subsetted reads for those between set start and end sites of a given chromosome. RPKB values were then averaged across 10 bp bins in a sliding window fashion of 1 bp step sizes (no padding was done to ensure all data used was real). For snRNAseq, position-sorted BAM files were converted into bigwig-type files via deepTools. These bigwig files were then read in to repeat the same type calculations as done for snATAC-seq, with read coverages also aggregated at a resolution of 10 bp and normalized via RPKM (Supplementary Fig. 9B). The map of the lentiviral Vector pRRL-MSCV-optNYESO optsr39TK-WPRE-1 used in the manufacturing of PBSC cell products is shown, which was used to alight the reads to in both A and B (Supplementary Fig. 9C).

### Reporting summary

Further information on research design is available in the Nature Portfolio Reporting Summary linked to this article.

## Data availability

Raw sequencing data are available via dbGaP (accession number phs003898.v1; https://www.ncbi.nlm.nih.gov/gap/advanced_search/?TERM=phs003898.v1). Source data for other figures are provided as a Source Data file. All other data (PET/CT images) are available upon request from the corresponding author only due to patient confidentiality laws. Source data are provided with this paper.

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

## Acknowledgements

This study was funded by the California Institute for Regenerative Medicine (CIRM) grant CLIN2-11380. T.S.N. was further supported by NIH grant K08 CA241088, Hyundai Hope on Wheels, the Tower Cancer Research Foundation. This study was additionally funded in part by the Parker Institute for Cancer Immunotherapy (PICI), NIH grant R35 CA197633 and the Ressler Family Fund (to A.R.). N.N.A. is supported by the Alan Ghitis Fellowship for Melanoma Research. N.F. is supported by the National Center for Advancing Translational Science (NCATS) of the National Institutes of Health under the UCLA Clinical and Translational Science Institute grant number UL1TR001881 and California Institute for Regenerative Medicine (CIRM) INFR4 Alpha Clinic Network Expansion Award. D.B.K. is supported by endowment funds from the UCLA Eli and Edyth Broad Stem Cell Research Center. The authors gratefully acknowledge the NHLBI National Gene Vector Biorepository at Indiana University (PI Cornetta, P40HL116212) for assistance in the generation and maintenance of the NY-ESO-1 TCR retrovirus and NY-ESO-1 TCR/sr39TK lentivirus vectors. Flow cytometry was performed in the UCLA Jonsson Comprehensive Cancer Center (JCCC) and Center for AIDS Research Flow Cytometry Core Facility that is supported by National Institutes of Health awards P30 CA016042 and 5P30 AI028697, and by the JCCC, the UCLA AIDS Institute, the UCLA Technology Center for Genomics and Bioinformatics (TCGB) core lab, and the David Geffen School of Medicine at UCLA. The authors wish to acknowledge the participation of patients and their caregivers, and the patient care at the Hematology-Oncology Stem Cell Transplantation Unit at UCLA.

## Author contributions

T.S.N. and A.R. served as principal investigators for the clinical trial and its design/execution. C.P-S., J.Z., D.B., L.Y., D.B.K., O.N.W., and A.R. conceived and developed the lentiviral vector and its implementation within the clinical protocol. T.S.N., N.N.A., C.W.P., and A.R. wrote the manuscript. I.B.C. and P.K-L. supervised clinical trial regulatory and compliance regulations. B.C-A., M.M., C.W.P., B.B-M., I.P.G., J.P., J.R., A.C., and A.V-C. conducted transgenic cell product manufacturing. T.S.N., A.S., B.C., N.F., G.M.S., S.J.L., and A.R. screened, enrolled, treated, and/or cared for patients on the clinical study protocol. B.C-A. and S.H. conducted flow cytometry analysis. N.N.A., C.W.P., M.K., and C.K. performed single-cell sequencing assays. N.N.A., E.M., and D.C. conducted the bioinformatics analysis. G.C. and M.A-A. designed, conducted, and interpreted the [18 F]-FHBG PET scans. C.W.P., M.K., and C.K. performed the antigen-dependent stimulation assays and ELISAs. A.M.K. assisted with statistical analysis and test design. All authors reviewed and edited the manuscript.

## Competing interests

TSN reports consulting honoraria from Allogene Therapeutics, PACT Pharma, Adaptive Biotechnologies, and Medidata Solutions. NF has received honoraria for speaker's bureaus and advisory boards from Bayer AG, Fennec Pharmaceuticals, and Springworks Therapeutics, and holds stock ownership in Moderna, Bolt Biotherapeutics, Regulus, Bluebird Bio, and 2seventy Bio. BC has received honoraria for consulting and advisory boards from Novartis, Delcath Systems, Instil Bio, Replimune, Atreca, Regeneron, Treeline Biosciences, and SpringWorks Therapeutics, and has received research funding from Bristol-Myers Squibb, Macrogenics, Karyopharm Therapeutics, Infinity Pharmaceuticals, Advenchen Laboratories, Xencor, Compugen, Iovance Biotherapeutics, RAPT Therapeutics, IDEAYA Biosciences, Ascentage Pharma, Atreca. Replimune, Instil Bio, Adagene, TriSalus Life Sciences, Kinnate Biopharma, PTC Therapeutics, Xilio Therapeutics, Kezar Life Sciences, Immunocore, AskGene Pharma. O.N.W. currently has consulting, equity, and/or board relationships with Trethera Corporation, Kronos Biosciences, Sofie Biosciences, Breakthrough Properties, Vida Ventures, Nammi Therapeutics, Two River, Iconovir, Appia BioSciences, Neogene Therapeutics, 76Bio, and Allogene Therapeutics. A.R. has received honoraria from consulting with Amgen and Roche-Genentech, is or has been a member of the scientific advisory board, and holds stock in Appia, Apricity, Arcus, Compugen, CytomX, ImaginAb, ImmPact, Inspirna, Kite-Gilead, Larkspur, Lutris, MapKure, Merus, Synthekine, and Tango, has received research funding from Agilent and from Bristol-Myers Squibb through Stand Up to Cancer (SU2C), and patent royalties from Arsenal Bio. None of these companies contributed to or directed any of the research reported in this article. D.B. and L.Y. are inventors on patents related to this study filed by California Institute of Technology. All other authors have no competing interests to declare.

## Additional information

[1]Division of Pediatric Hematology-Oncology, Department of Pediatrics, University of California Los Angeles, Los Angeles, CA, USA. [2]Department of Microbiology, Immunology, and Molecular Genetics, University of California Los Angeles, Los Angeles, CA, USA. [3]Jonsson Comprehensive Cancer Center, University of California Los Angeles, Los Angeles, CA, USA. [4]Eli and Edythe Broad Center for Regenerative Medicine and Stem Cell Research, University of California Los Angeles, Los Angeles, CA, USA. [5]Molecular Biology Institute, University of California Los Angeles, Los Angeles, CA, USA. [6]David Geffen School of Medicine, University of California Los Angeles, Los Angeles, CA, USA. [7]Division of Hematology-Oncology, Department of Medicine, University of California Los Angeles, Los Angeles, CA, USA. [8]Ahmanson Translational Theranostics Division, Department of Molecular and Medical Pharmacology, University of California Los Angeles, Los Angeles, CA, USA. [9]Division of Surgical Oncology, Department of Surgery, University of California Los Angeles, Los Angeles, CA, USA. [10]Parker Institute for Cancer Immunotherapy, UCLA, Los Angeles, CA, USA. [11]Department of Molecular and Medical Pharmacology, University of California Los Angeles, Los Angeles, CA, USA. [12]Department of Orthopaedic Surgery, University of California Los Angeles, Los Angeles, CA, USA. [13]Department of Medicine Statistics Core, David Geffen School of Medicine, University of California Los Angeles, Los Angeles, CA, USA. [14]Division of Biology and Biological Engineering, California Institute of Technology, Pasadena, CA, USA. ✉e-mail: tnowicki@mednet.ucla.edu

