## [Transparent Peer Review file · Nature Communications]

Human Cancer-Targeted Immunity via Transgenic Hematopoietic Stem Cell Progeny

Corresponding Author: Dr Theodore Nowicki

Version 0:

Reviewer comments:

Reviewer #1

(Remarks to the Author)

In this manuscript, the authors have reported the results of a FIH phase I clinical trial using tandem cell therapies involving conventional T cells and HSCs both of which are engineered to express the NY-ESO-1-specific TCR for the treatment of NY-ESO-1+ solid tumors. I agree with the rationale of this study aimed at enhancing the efficacy of immunotherapy by perpetuating the survival of antigen-specific T cells in the body. Given the small number of patients who completed treatment, and the fact that one patient died from de novo CMV infection, there seems to be serious safety issues associated with this study. However, I think that this unprecedented research endeavor is of great significance and importance. The work has potential far-reaching implications and may particularly impact the cell therapy field.

Major concerns

1, I believe that it is significantly important in this study to show the participation of T cells, especially those derived from the infused HSCs, in the anti-tumor immune responses. How about the presence or absence of pLenti-positive T cells in the sampled tumor tissues, especially in the case of NYSCT-03? If it is not possible in the protocol, the authors have to state the reasons in this manuscript.

2, As described in the manuscript, while sr39TK has advantages, its immunogenicity may lead to accelerated rejection of HSCs-derived cells. Have the authors analyzed T cell immune responses specific to sr39TK by using patients' peripheral blood? If so, these data could help us to understand why engraftment in NYSCT-05 differs from that in NY-SCT-03.

3, More detailed results should be presented on the gene expression profile between Plenti-positive CD8 T cells derived from HSCs and Plenti-negative CD8 T cells. Do Plenti positive cells express a similar gene profile compared with Plenti-negative CD8 T cells? Were there any differences in the expression of genes related to anergy, such as GRAIL? What about exhaustion-related gene expressions? Although the authors suggest that HSC-derived cells are functional in terms of cytokine production, these data would strengthen this inference.

Reviewer #2

(Remarks to the Author)

The work from Nowicki T et al. entitled "Human Cancer-Targeted Immunity via Transgenic Hematopoietic Stem Cell Progeny" presents interesting results regarding the advantages of a double infusion of TCR-engineered T and PBSC (peripheral blood stem cells) cellular products for the treatment of solid malignancies (sarcoma). Out of five recruited patients, two were excluded from the study and did not receive TCR-engineered PBSC. The authors present results of the three remaining patients, with two cases of clinical response.

The strongest asset of this study is clearly its novelty, as at the best of my knowledge it is the first report employing this strategy in solid tumor patients. Unfortunately, I am not persuaded by the therapeutic benefit of the approach in respect to its complexity, impact on the normal functioning of immune cells and genotoxic risk that stem cell modification entails. The goal of increasing fitness and persistence of engineered products is the subject of many successful approaches that do not involve modification of stem cells but simply focus on T cells (manufacturing optimization, further genetic manipulation to

overexpress or knock-out relevant genes).

More specifically, I am concerned that the proposed approach results in the presence of the exogenous TCR in all immune cell population; even if you consider that CD3 is required for TCR expression on cell surface, this would imply constitutive expression on T cells but also on Tregs, NK cells etc. Potential impacts on physiological immunity (i.e. against viral pathogens) are not clear, especially considering that the authors observe some incomplete degree of allelic exclusion demonstrated by co-expression of transgenic and genomic TCRs within the T cell cluster. Clinical responses observed in only two patients are not quite sufficient to support the claim that the approach does not inherently carry important risks, considering that long-term persistence was nevertheless not achieved. Translating this approach to a CAR-based strategy, as proposed in the discussion, may conversely increase the risks, since constitutive CAR expression on all immune cells would be achieved. In addition, it is known in literature that HSC-derived anti-CD19 CAR-T cells cannot survive thymic selection (e.g. PMID: 19033646). Could this be one of the mechanisms behind the lack of persistence of PBSC-derived cells? In general, I would have considered more positively an inducible system able to drive transgene expression only on mature T cells.

In addition to the point above, I find that there are some issues with the study that prevent from drawing significant conclusions.

- The number of patients is an important limiting factor in assessing the significance of the results. NYSCT-01 was affected from a different kind of sarcoma in respect to the other four patients with a completely different prior treatment strategy, which may greatly affect T cell populations and responses. Two patients were excluded from the study due to the fact that PBSC engineering was not feasible. Overall, conclusions are drawn from two patients, which is not sufficient to deliver any certainty on the results.
- Regarding safety of the therapy, the authors claim that no serious adverse event may be attributed to TCR-engineered cellular products, with only one case that may be related to retroviral modified NYESO-1 TCR PBMC.
 - o A more in-depth explanation on how they would distinguish adverse events from the two cellular product is needed.
 - o I find that Extended table 2 does not offer objective parameters to assess observed adverse events: who do the authors distinguish events “definitely”, “probably” and “possibly” related to conditioning regimens, LV-NYESO TCR/sr39TK PBSC, IL-2 and other variables.
 - o Data presentation in different tables make it difficult to understand which patient experienced which adverse event. Do the authors find a correlation of increased adverse events in the two patients that achieve tumor response? It is difficult to discern whether the therapy can be considered safe in itself or whether it was due to underperforming efficacy profile.
- The authors report lack of expansion of transgenic TCR-T in NYSCT-01 (ascribing this effect to a suboptimal lymphodepletion with fludarabine) and “superior peripheral lymphodepletion and subsequent in vivo expansion of the transgenic TCR-T” in NYSCT-03 and NYSCT-05. This increased in vivo expansion cannot be appreciated in Extended Figure 4, especially in patient 05.
- NYSCT-05 show continuous increase of VCN in LVV-transduced PBSC-derived T cellular products. Could this represent a safety concern?

Reviewer #3

(Remarks to the Author)

Please check if all names are written in full before using abbreviations.

The abstract provides limited information on the results; it would be helpful to report the findings with a bit more detail to make the abstract more descriptive.

In the introduction, it does not specify the type of cancer being treated or why the NY-ESO-1 target was chosen. It would be beneficial to describe the target in more detail and to provide data on the expression percentage of this protein in tumors of the same type.

It may also be beneficial to mention that the FDA recently approved the first treatment with transgenic TCR-T cells, which would lend support to the choice of therapy.

Clarify the rationale for conducting tandem therapy with autologous T cells and autologous hematopoietic stem cells (HSCs): What is the role of each subpopulation, and what benefit might their combination provide?

Results:

CD34+ cells were selected, but was there no selection of T cells from the PBMCs?

Was the transduction done on general PBMCs?

Were the PBMCs injected fresh or frozen like the PBSCs?

Table 1: Ensure font size consistency, keep entire words on the same line, and specify which patients were withdrawn from the study.

Figure 1: The timeline appears somewhat unclear; a linear presentation or dividing it into stages may improve clarity.

In line 146, the statement is unclear. It mentions no grade 3 adverse events associated with HSCs but does with TCR. If this is a combined therapy, why was this distinction made?

Extended Data Figure 3 appears misaligned, with text overlapping the charts.

Extended Data Table 3: Were only these points analyzed for each patient? Were patients with a single analysis included?

Line 198: Cite Figures 2A and 2B before 2C and 2D or rearrange the figure order to match the citation sequence. (This applies to Figure 3 as well.)

Was the functional assay performed on HSCs from all patients? It would be beneficial to describe results for all patients to assess potential tumor remission capabilities. Additionally, functional testing of retrovirus-modified T cells would be valuable.

I suggest assays to evaluate whether both cell subpopulations are indeed necessary, as it appears that HSCs alone might be more functional. It would be interesting to place both subpopulations in vitro with the tumor, assessing proliferation, cytotoxicity, and activation/exhaustion markers, while comparing this to cultures with only HSCs or PBMCs. This would provide insight into the functional role of each subpopulation and any potential synergistic effects.

If additional cells are available, an in vivo experiment with controls using only HSCs, modified T cells, and tandem therapy as another arm, would also be insightful.

The results are promising, and it might be helpful to further explore these findings to better understand bottlenecks and optimize experiments.

Another interesting aspect would be to assess the functionality of non-T cells that also expressed the TCR against NY-ESO-1. Are these cells functional as well? Performing functional experiments with T cells derived from HSCs alone or with the general PBMC population could provide insight into whether the broader PBMC population contributes to the overall activity.

Discussion:

Discuss the effects of other modified non-T cell populations derived from HSCs.

Compare these findings with similar results in the literature targeting the same antigen using transgenic TCR therapy.

Explore the rationale for using a tandem approach with T cells and HSC-derived cells rather than using each subpopulation in isolation.

Reviewer #4

(Remarks to the Author)

Version 1:

Reviewer comments:

Reviewer #1

(Remarks to the Author)

The authors have addressed all my questions sufficiently.

Reviewer #2

(Remarks to the Author)

The authors made an effort to address raised points, even though lack of patients' samples availability restricted the possibility of including additional experimental data. Even though dissemination of these novel results may likely be of impact in the field, I reckon that some discussion could be better explored to support the reader's understanding of the limitations of the study, word count restrictions notwithstanding.

For example, I am still not persuaded that the safety of the approach is proven by presented results. Indeed, besides toxicity issues discussed by the authors after revisions, the long-term impact of the approach, e.g. the consequences of partial allelic exclusion, couldn't be assessed due to lack of persistence. I agree that the use of a CAR construct might avoid the partial allelic exclusion phenomenon, but this in turn raises several limitations that should be better discussed, linked to CAR expression on progeny of transduced HSPC and the relative unknown impact of this on their functioning.

In terms of efficacy, lack of persistence negatively impacts one of the intrinsic advantages of the approach, with no clear explanation on the cause of this effect.

Giving the elevated risks of potential toxicities associated with this approach, including on-target off-tumor potential in case of antigens also expressed in healthy tissues (albeit at low levels), I would encourage the authors to better discuss this point, making a statement on the relevance of including a suicide gene or other systems to restrict receptor expression to a specific cell type or timeframe.

Minor: Figure 4B is duplicated unnecessary in Figure 5A.

Reviewer #3

(Remarks to the Author)

I consider that the authors' responses to the questions raised were adequate and the manuscript should be accepted for publication.

Reviewer #4

(Remarks to the Author)

Reviewer #1 (Remarks to the Author):

In this manuscript, the authors have reported the results of a FIH phase I clinical trial using tandem cell therapies involving conventional T cells and HSCs both of which are engineered to express the NY-ESO-1-specific TCR for the treatment of NY-ESO-1+ solid tumors. I agree with the rationale of this study aimed at enhancing the efficacy of immunotherapy by perpetuating the survival of antigen-specific T cells in the body. Given the small number of patients who completed treatment, and the fact that one patient died from de novo CMV infection, there seems to be serious safety issues associated with this study. However, I think that this unprecedented research endeavor is of great significance and importance. The work has potential far-reaching implications and may particularly impact the cell therapy field.

Response: *We thank the reviewer for their positive assessment of our work and its impact in the cell therapy field.*

Major concerns

1, I believe that it is significantly important in this study to show the participation of T cells, especially those derived from the infused HSCs, in the anti-tumor immune responses. How about the presence or absence of pLenti-positive T cells in the sampled tumor tissues, especially in the case of NYSCT-03? If it is not possible in the protocol, the authors have to state the reasons in this manuscript.

Response: *Unfortunately, we were unable to obtain any on-treatment tumor biopsy samples in order to examine this important question, owing to both the clinical instability of subject NYSCT-03 as well as the COVID19 pandemic at the time limiting the ability of us to pursue elective procedures for such tumor biopsy samples. We have stated these limitations, as well as their reasons, in the manuscript text per the reviewer's request.*

2, As described in the manuscript, while sr39TK has advantages, its immunogenicity may lead to accelerated rejection of HSCs-derived cells. Have the authors analyzed T cell immune responses specific to sr39TK by using patients' peripheral blood? If so, these data could help us to understand why engraftment in NYSCT-05 differs from that in NY-SCT-03.

Response: *We have included new experimental data to address this question, the results of which are presented in **Extended Data Figure 5**. In brief, we co-cultured subjects' untransduced or transduced PBMCs with both untransduced and transduced PBSCs, the latter of which were transduced with both the NY-ESO-1 TCR and sr39tk. We then assayed the co-culture media via interferon-gamma ELISA, as we had done for the lot release of the TCR-T cells to confirm reactivity to NY-ESO-1. We observed no significant increase in interferon-gamma release in patient-matched transduced PBSCs relative to the untransduced PBSCs, consistent with a lack of immunogenicity of sr39tk as a causative mechanism for rejection of the transgenic PBSCs. In the case of subject NYSCT-05, the relatively low degree of transduction of the PBSC products may have simply resulted in poor persistence of the transgenic PBSCs. We acknowledge this point, as well as the fact that our in vitro work does not fully recapitulate the complexity of the bone marrow niche in vivo, in the manuscript text.*

3, More detailed results should be presented on the gene expression profile between Plenti-positive CD8 T cells derived from HSCs and Plenti-negative CD8 T cells. Do Plenti positive cells express a similar gene profile compared with Plenti-negative CD8 T cells? Were there any differences in the expression of genes related to anergy, such as GRAIL? What about

exhaustion-related gene expressions? Although the authors suggest that HSC-derived cells are functional in terms of cytokine production, these data would strengthen this inference.

Response: *We have conducted additional analysis on the single-cell transcriptomic data obtained from NYSCT-03. Of note, focusing on the pLenti+ and pLenti- CD8 Tcell populations showed no difference in energy and exhaustion markers expression in the CD8 Tcells across the pLenti+ and pLenti- cells (Extended Data Figure 6 A&B). Looking into the top 100 differentially expressed genes between pLenti+ and pLenti- CD8 Tcells did not reveal key transcriptional differences between the two CD8 Tcells populations. Although CD80 appeared to be upregulated in a few pLenti+ CD8 Tcells (only 3 out of 24 cells) as compared to the pLenti- CD8 Tcells, we suspect that the pLenti+ CD8 Tcells are new progeny that wouldn't have had the chance to undergo thymic selection (Extended Data Figure 6 C).*

Reviewer #2 (Remarks to the Author):

The work from Nowicki T et al. entitled "Human Cancer-Targeted Immunity via Transgenic Hematopoietic Stem Cell Progeny" presents interesting results regarding the advantages of a double infusion of TCR-engineered T and PBSC (peripheral blood stem cells) cellular products for the treatment of solid malignancies (sarcoma). Out of five recruited patients, two were excluded from the study and did not receive TCR-engineered PBSC. The authors present results of the three remaining patients, with two cases of clinical response.

The strongest asset of this study is clearly its novelty, as at the best of my knowledge it is the first report employing this strategy in solid tumor patients. Unfortunately, I am not persuaded by the therapeutic benefit of the approach in respect to its complexity, impact on the normal functioning of immune cells and genotoxic risk that stem cell modification entails. The goal of increasing fitness and persistence of engineered products is the subject of many successful approaches that do not involve modification of stem cells but simply focus on T cells (manufacturing optimization, further genetic manipulation to overexpress or knock-out relevant genes).

Response: *We acknowledge that the ultimate therapeutic potential of this approach is limited in this cancer type and patient population, owing to their overall medical fragility by the time they would have failed standard of care and been in the phase 1 trial space, and have expanded upon this point in the discussion section.*

More specifically, I am concerned that the proposed approach results in the presence of the exogenous TCR in all immune cell population; even if you consider that CD3 is required for TCR expression on cell surface, this would imply constitutive expression on T cells but also on Tregs, NK cells etc. Potential impacts on physiological immunity (i.e. against viral pathogens) are not clear, especially considering that the authors observe some incomplete degree of allelic exclusion demonstrated by co-expression of transgenic and genomic TCRs within the T cell cluster. Clinical responses observed in only two patients are not quite sufficient to support the claim that the approach does not inherently carry important risks, considering that long-term persistence was nevertheless not achieved. Translating this approach to a CAR-based strategy, as proposed in the discussion, may conversely increase the risks, since constitutive CAR expression on all immune cells would be achieved. In addition, it is known in literature that HSC-derived anti-CD19 CAR-T cells cannot survive thymic selection (e.g. PMID: 19033646). Could

this be one of the mechanisms behind the lack of persistence of PBSC-derived cells? In general, I would have considered more positively an inducible system able to drive transgene expression only on mature T cells.

Response: *While the difficulty of CAR-T cells in surviving thymic selection has been reported, in the case of both our subjects reported here and our previously reported experiences with NY-ESO-1 TCR-T cell clinical trials (Nowicki et al, Clinical Cancer Research 2019, Frankiw et al J Immunother Cancer 2023), most subjects have detectable endogenous NY-ESO-1 TCR clones detectable in circulation prior to TCR transduction, implying that NY-ESO-1 is sufficiently antigenic as to generate TCR clones which survive thymic selection (albeit insufficiently to generate a sufficiently strong clonal proliferation in itself). Given that we are using such a TCR as is present in natural endogenous settings, we would speculate that this would be an advantage in using a TCR as opposed to a CAR. The PBSC-derived cells observed in our study did not display persistence due to the patient's unfortunate death due to opportunistic infection, and it is not possible to infer anything further regarding their persistence. However, this limitation due to the subject's death, as well as our previously documented presence of endogenous NY-ESO-1 TCR clones, as well as those within our present study, is further expanded upon in the results and discussion sections.*

In addition to the point above, I find that there are some issues with the study that prevent from drawing significant conclusions.

- The number of patients is an important limiting factor in assessing the significance of the results. NYSCT-01 was affected from a different kind of sarcoma in respect to the other four patients with a completely different prior treatment strategy, which may greatly affect T cell populations and responses. Two patients were excluded from the study due to the fact that PBSC engineering was not feasible. Overall, conclusions are drawn from two patients, which is not sufficient to deliver any certainty on the results.

Response: *As discussed above, we acknowledge that the ultimate clinical utility of this approach is not likely to be compelling in this setting. We further appreciate that the heterogeneity of sarcomas here may further limit this as well, and have added text in the manuscript to emphasize this point.*

- Regarding safety of the therapy, the authors claim that no serious adverse event may be attributed to TCR-engineered cellular products, with only one case that may be related to retroviral modified NYESO-1 TCR PBMC.

- o A more in-depth explanation on how they would distinguish adverse events from the two cellular product is needed.

- o I find that Extended table 2 does not offer objective parameters to assess observed adverse events: how do the authors distinguish events “definitely”, “probably” and “possibly” related to conditioning regimens, LV-NYESO TCR/sr39TK PBSC, IL-2 and other variables.

- o Data presentation in different tables make it difficult to understand which patient experienced which adverse event. Do the authors find a correlation of increased adverse events in the two patients that achieve tumor response? It is difficult to discern whether the therapy can be considered safe in itself or whether it was due to underperforming efficacy profile.

Response: *The manner in which all of the adverse events are characterized and presented are consistent with all CTCAE criteria for all FDA reporting, and are standardized in all manuscripts reporting clinical trial data for both cell therapy products and other clinical interventions, including all previous manuscripts from our group and others. In the case of the different products, the reported adverse events are typically distinguished by their proximity from*

administration of a given agent. In the case of the grade 3 hypotension that is listed as “possibly” related to the PBMC product, the event happened within 24-48 hours following PBMC administration in the setting of cytokine release syndrome, which we thus attributed to the PBMC product, as this would be consistent with other previously reported adverse event profiles in both TCR-T and CAR-T cell therapy products.

- The authors report lack of expansion of transgenic TCR-T in NYSCT-01 (ascribing this effect to a suboptimal lymphodepletion with fludarabine) and “superior peripheral lymphodepletion and subsequent in vivo expansion of the transgenic TCR-T” in NYSCT-03 and NYSCT-05. This increased in vivo expansion cannot be appreciated in Extended Figure 4, especially in patient 05.

Response: We apologize for any confusion on this point. The initial in vivo expansion to which we refer is that of the TCR-T cells themselves, immediately following TCR-T cell infusion (within the first 2-3 weeks post-infusion), and this is demonstrated in the y-axes of Figure 2-4, whereas the data shown in Extended Figure 4 is simply the degree of total lymphodepletion. The in vivo expansion would only be reflected in the TCR+ T cells, and not the total lymphocyte counts (which were only included to demonstrate the degree of lymphodepletion from the conditioning chemotherapy). We have edited the manuscript text in order to further clarify this point.

- NYSCT-05 show continuous increase of VCN in LVV-transduced PBSC-derived T cellular products. Could this represent a safety concern?

Response: The data in question were for subject NYSCT-03, and no such data are presented for NYSCT-05, who actually demonstrated no persistence of any PBSC-derived products. In the case of NYSCT-03, we hypothesize that the PBSC-derived products would have equilibrated at 3-6 months post-transplant, consistent with what has previously been reported in the setting of autologous HSCT immune reconstitution. As the subject unfortunately passed away due to opportunistic infection, it is unknown at what point the PBSC-derived products would have equilibrated. However, this limitation has been expanded upon in the discussion section.

Reviewer #3 (Remarks to the Author):

Please check if all names are written in full before using abbreviations.

Response: This has been done.

The abstract provides limited information on the results; it would be helpful to report the findings with a bit more detail to make the abstract more descriptive.

Response: We were attempting to both stay within the 200 word count limit for the abstract, as well as conform to the Nature style guide for abstracts, which unfortunately limits our ability to discuss the results in further detail in the abstract. We chose to highlight the salient features of our study in regards to both safety/feasibility of the phase 1 trial, and the fact that the lentiviral-derived T-cell progeny display significant and specific antitumor activity, which is the first time such a result has been reported in human subjects. We have, however, expanded the abstract to include that the T-cell progeny did not display evidence of anergy or exhaustion.

In the introduction, it does not specify the type of cancer being treated or why the NY-ESO-1

target was chosen. It would be beneficial to describe the target in more detail and to provide data on the expression percentage of this protein in tumors of the same type.

Response: *As NY-ESO-1 is a pan-cancer antigen, we sought to recruit any patients with advanced solid malignancies, as stated in the introduction. We have modified the text to emphasize that NY-ESO-1 is expressed in a wide variety of malignancies, predominantly sarcomas, and that it is not generally expressed in normal somatic tissues, which is the primary impetus for all NY-ESO-1 targeting therapeutics.*

It may also be beneficial to mention that the FDA recently approved the first treatment with transgenic TCR-T cells, which would lend support to the choice of therapy.

Response: *The FDA approval of Tecelra had not occurred by the time of this manuscript's initial submission, and was not approved during the actual conduct of the trial (2018-2022). However, we have added this point within the discussion section.*

Clarify the rationale for conducting tandem therapy with autologous T cells and autologous hematopoietic stem cells (HSCs): What is the role of each subpopulation, and what benefit might their combination provide?

Response: *The introduction section already states this: "[NY-ESO-1 TCR-T cells] provided an initial wave of antitumor activity arising from the traditional TCR transduced T cell product, after which the T cell progeny from the HSC niche were shown to provide circulating transgenic NY-ESO-1 TCR T cells which displayed specific NY-ESO-1 antigen-dependent antitumor functionality"*

Results:

CD34+ cells were selected, but was there no selection of T cells from the PBMCs?
Was the transduction done on general PBMCs?

Response: *Yes, we have clarified this point in the manuscript text.*

Were the PBMCs injected fresh or frozen like the PBSCs?

Response: *PBMCs were injected fresh; this is already stated in the manuscript text.*

Table 1: Ensure font size consistency, keep entire words on the same line, and specify which patients were withdrawn from the study.

Response: *This has been done.*

Figure 1: The timeline appears somewhat unclear; a linear presentation or dividing it into stages may improve clarity.

Response: *This has been done.*

In line 146, the statement is unclear. It mentions no grade 3 adverse events associated with HSCs but does with TCR. If this is a combined therapy, why was this distinction made?

Response: *As stated in the replies to other reviewer critiques above, in the case of the different products, the reported adverse events are typically distinguished by their proximity from*

administration of a given agent. In the case of the grade 3 hypotension that is listed as “possibly” related to the PBMC product, the event happened within 24-48 hours following PBMC administration in the setting of cytokine release syndrome, which we thus attributed to the PBMC product, as this would be consistent with other previously reported adverse event profiles in both TCR-T and CAR-T cell therapy products.

Extended Data Figure 3 appears misaligned, with text overlapping the charts.

Response: *This does not appear to be an issue when viewed from our end. However, we include source files for all components of the manuscript, so it is possible that there was an issue with the pdf conversion on this reviewer’s end. We have provided source word file documents of all tables, including this one, so that any formatting errors can be removed if they occur downstream of our submission.*

Extended Data Table 3: Were only these points analyzed for each patient? Were patients with a single analysis included?

Response: *We only included data points for the day +30 and day +120 scans, which are the only timepoints indicated in our clinical protocol. Subject NYSCT-03 passed away prior to the day +120 timepoint. Subject NYSCT-05 displayed no engraftment at day +30, so no day +120 scan was performed in his case. The manuscript text has been modified to emphasize these points.*

Line 198: Cite Figures 2A and 2B before 2C and 2D or rearrange the figure order to match the citation sequence. (This applies to Figure 3 as well.)

Response: *This has been done.*

Was the functional assay performed on HSCs from all patients? It would be beneficial to describe results for all patients to assess potential tumor remission capabilities. Additionally, functional testing of retrovirus-modified T cells would be valuable.

Response: *The functional assay (antigen-dependent interferon-gamma release) was not performed on HSCs, as these cell types are known to not produce this cytokine until subsequent differentiation into terminally differentiated lymphocytes. Functional testing was performed on the retrovirus-modified T-cells as part of the lot release criteria, and these data are now summarized in **Extended Data Table 2 and 6**.*

I suggest assays to evaluate whether both cell subpopulations are indeed necessary, as it appears that HSCs alone might be more functional. It would be interesting to place both subpopulations in vitro with the tumor, assessing proliferation, cytotoxicity, and activation/exhaustion markers, while comparing this to cultures with only HSCs or PBMCs. This would provide insight into the functional role of each subpopulation and any potential synergistic effects. If additional cells are available, an in vivo experiment with controls using only HSCs, modified T cells, and tandem therapy as another arm, would also be insightful. The results are promising, and it might be helpful to further explore these findings to better understand bottlenecks and optimize experiments.

Response: *While we acknowledge the potential utility of such experiments, we used up all remaining cells in order to perform the requested HSC/PBMC co-culture experiments requested*

by reviewer 1 in order to demonstrate lack of sr39tk-driven HSC rejection. We have acknowledged this limitation within the manuscript text.

Another interesting aspect would be to assess the functionality of non-T cells that also expressed the TCR against NY-ESO-1. Are these cells functional as well? Performing functional experiments with T cells derived from HSCs alone or with the general PBMC population could provide insight into whether the broader PBMC population contributes to the overall activity.

Response: *While our scRNAseq data shows the presence of the transgene in non-T cell populations, it is highly unlikely that any functional protein would lead to antigen specific activity of, for example, a macrophage, given that these cells would lack the other underlying intracellular machinery necessary to effectuate a TCR-driven response. We have added text addressing this to the manuscript.*

Discussion:

Discuss the effects of other modified non-T cell populations derived from HSCs.

Response: *We have added this to the discussion section.*

Compare these findings with similar results in the literature targeting the same antigen using transgenic TCR therapy.

Response: *We have expanded upon this point in the discussion section.*

Explore the rationale for using a tandem approach with T cells and HSC-derived cells rather than using each subpopulation in isolation.

Response: *This text is already present in both the introduction and discussion sections, but we have expanded upon it.*

Reviewer #4 (Remarks to the Author):

Response: *We thank the reviewer for their efforts.*

Reviewer #2 (Remarks to the Author):

The authors made an effort to address raised points, even though lack of patients' samples availability restricted the possibility of including additional experimental data. Even though dissemination of these novel results may likely be of impact in the field, I reckon that some discussion could be better explored to support the reader's understanding of the limitations of the study, word count restrictions notwithstanding.

For example, I am still not persuaded that the safety of the approach is proven by presented results. Indeed, besides toxicity issues discussed by the authors after revisions, the long-term impact of the approach, e.g. the consequences of partial allelic exclusion, couldn't be assessed due to lack of persistence. I agree that the use of a CAR construct might avoid the partial allelic exclusion phenomenon, but this in turn raises several limitations that should be better discussed, linked to CAR expression on progeny of transduced HSPC and the relative unknown impact of this on their functioning.

In terms of efficacy, lack of persistence negatively impacts one of the intrinsic advantages of the approach, with no clear explanation on the cause of this effect.

Giving the elevated risks of potential toxicities associated with this approach, including on-target off-tumor potential in case of antigens also expressed in healthy tissues (albeit at low levels), I would encourage the authors to better discuss this point, making a statement on the relevance of including a suicide gene or other systems to restrict receptor expression to a specific cell type or timeframe.

Response: *We have expanded the text in the introduction and discussion sections to highlight the fact that we did indeed include a suicide gene in our construct. We have also expanded the text to highlight the reviewer's point on the unknown aspect of CAR transduction at the HSC level and how that underscores the need for continued use of such safety features as suicide genes.*

As we have previously stated, the lack of persistence in our case was largely due to the clinical fragility of the subjects treated. We highlight the derivation of functional TCR-T cell progeny from the lentiviral niche, as well as the previously reported long-term stability of lentiviral-transduced HSC products, to speculate that this may be possible in other clinical contexts.

Minor: Figure 4B is duplicated unnecessary in Figure 5A.

Response: *We felt that inclusion of this panel in Figure 5A, as well as highlighting the timepoint for downstream analysis, was most helpful for the reader in terms of understanding the remainder of this figure.*